# ACTIVE LEARNING WITH CONTROLLABLE AUGMENTATION INDUCED ACQUISITION

## ABSTRACT

The mission of active learning is to iteratively identify the most *informative* data samples to annotate, and therefore to attain decent performance with much fewer samples. Despite the promise, the acquisition of informative unlabeled samples can be unreliable — particularly during early cycles — owning to limited data samples and sparse supervision. To tackle this, the data augmentation techniques seem straightforward yet promising to easily extend the exploration of the input space. In this work, we thoroughly study the coupling of data augmentation and active learning whereby we propose **C**ontrollable **A**ugmentation **M**ani**P**ulator for **A**ctive **L**earning. In contrast to the few prior work that touched on this line, CAMPAL emphasizes a tighten and better-controlled integration of data augmentation into the active learning framework, as in three folds: (i)-carefully designed data augmentation policies applied separately on labeled and unlabeled data pool in every cycle; (ii)-controlled and quantifiably optimizable augmentation strengths; (iii)-full but flexible coverage for most (if not all) active learning schemes. Through extensive empirical experiments, we bring the performance of active learning methods to a new level: an absolute performance boost of **16.99%** on CIFAR-10 and **12.25%** on SVHN with 1,000 annotated samples. Complementary to the empirical results, we further provide theoretical analysis and justification of CAMPAL.

## 1 INTRODUCTION

The acquisition of labeled data serves as a foundation for the remarkable successes of deep supervised learning over the last decade, which also incurs great monetary and time costs. *Active learning* (AL) is a pivotal learning paradigm that puts the data acquisition process into the loop of learning, locating the most informative and valuable data samples for annotation (Settles, 2009; Zhang et al., 2020; Kim et al., 2021a; Wu et al., 2021). With much-lowered sample complexity but comparable performance compared to its supervised counterpart, active learning is widely used in real-world applications and ML productions (Bhattacharjee et al., 2017; Feng et al., 2019; Hussein et al., 2016). In spite of its meritorious practicality, active learning often suffers from unreliable data acquisition, especially from the early stages. Notably, the models obtained around the early stages are generally raw and undeveloped due to the insufficient data curated and sparse supervision signal being consumed. The subsequent cycle of the data query is based on the model produced from the current cycle.

While this problem can probably be mitigated after adequate cycles are conducted, we argue that the problems at the early stages of AL cannot be overlooked. Indeed, few works have resorted to data augmentation techniques to generate additional data examples for data distribution enrichment, e.g. GAN-based (Tran et al., 2019) and STN-based (Kim et al., 2021b) methods. In this work, we attempt to take a further step in investigating the role of data augmentation for AL.

To begin with, we provide a straightforward quantitative observation in Figure 1. The setup of these results is rather simple: we directly apply vanilla DA operations, such as flipping and rotation, to data samples and linearly increase the augmentation strengths. We may conclude from these scores as follows. First, the simple augmentation (loosely) integrated into AL has led to surprisingly enhanced results, albeit their complicated designs. Secondly and perhaps more important, we have a counterfactual observation that the same augmentation policy facilitated on different data pools manifests notably different impacts. As shown in Figure 1, when gradually stacking the augmentation operations, the labeled and unlabeled data pools achieves the best performance at different levels of

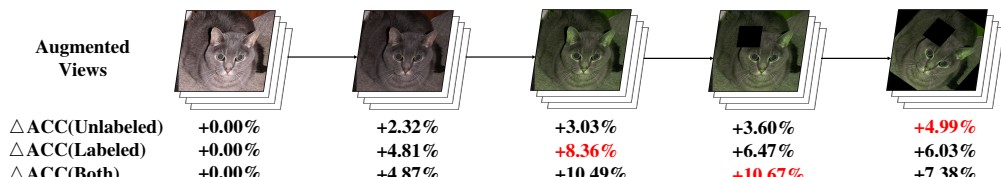

| Augmented Views | | | | | |
|---|---|---|---|---|---|
| △ACC(Unlabeled) | +0.00% | +2.32% | +3.03% | +3.60% | **+4.99%** |
| △ACC(Labeled) | +0.00% | +4.81% | **+8.36%** | +6.47% | +6.03% |
| △ACC(Both) | +0.00% | +4.87% | +10.49% | **+10.67%** | +7.38% |

Figure 1: A visualization for data augmentation and their corresponding performance change as we stack augmentations over images when integrating them into active learning cycles. We test 3 cases where augmentations are applied to 1) unlabeled samples only; 2) labeled samples only; 3) Both. Details of experimental setups can be found in Appendix B.2.

augmentation strengths. In hindsight, we hereby post our reasoning. To fully and tightly incorporate DA into AL schemes, the augmentation ought to serve different objectives on the labeled and unlabeled pools. In particular, the labeled pool favors label-preserving augmentation in order to obtain a strong and reliable classifier. By contrast, the unlabeled pool may require relatively more aggressive augmentation so as to maximally gauge the unexplored distribution. The phenomenon in Figure 1 preliminarily validates this reasoning. Noted, this counterfactual observation has not been studied or investigated by prior works (Tran et al., 2019; Gao et al., 2020; Kim et al., 2021b).

Motivated by it, we propose **C**ontrollable **A**ugmentation **M**ani**P**ulator for **A**ctive **L**earning. Core to our method is a purposely designed form of better controlled and tightened the integration of data augmentation into active learning. By proposing CAMPAL, we aim to fill this integration gap and unlock the full potential of data augmentation methods integrated into active learning schemes. In particular, CAMPAL integrates several mechanisms into the whole AL framework:

- CAMPAL constructs separate augmentation flows distinctly on labeled and unlabeled data pools towards their own objectives;
- CAMPAL composes a *strength* optimization procedure for the applied augmentation policies;
- CAMPAL complies with most common active learning schemes, with carefully designed acquisition functions for both score- and representation-based methods.

Besides the theoretical justification of CAMPAL offered in Section 4, we extensively conduct wide experiments and analyses on our approach. The empirical results of CAMPAL are stunning: a **16.99%** absolute improvement at a 1,000-sample cycle and a **13.34%** lead with 2,000 samples on CIFAR-10, compared with previously best methods. Arguably, we may postulate that these significantly enhanced results may have the chance to greatly extend the boundary of active learning research.

## 2 METHODOLOGY

In this section, we describe CAMPAL in detail. CAMPAL is chiefly composed of two components. On one hand, CAMPAL formulates a decoupled optimization workflow to locate feasible augmentations being applied to labeled/unlabeled data pools with distinct optimization objectives. This optimization difference is eventually manifested by their augmentation strength (Section 2.2). On the other hand, CAMPAL aggregates the information provided by properly-controlled augmentations with modified acquisition functions (Section 2.3), so as to be adaptable with most (if not all) active learning schemes. Hence we may posit that CAMPAL forms a much more tightened integration of DA and AL, due to not only its controllable mechanism on both data pools but also its full adaptability for all common active learning schemes. The framework for CAMPAL is summarized in Figure 2.

### 2.1 SETUP AND DEFINITIONS

**Active Learning.** The problem of active learning (AL) is defined with the following setup. Consider $\mathcal{D} \subset \mathbb{R}^d$ as the underlying dataset consisting of a labeled data pool $\mathcal{D}_L$ and an unlabeled data pool $\mathcal{D}_U$, with $|\mathcal{D}_U| \gg |\mathcal{D}_L|$. Based on a fully-trained classifier $f_\theta$ that assigns a label to each data point, a data acquisition function $h_{acq}(x, f_\theta) : \mathcal{D}_U \to \mathbb{R}$ calculates the score for each data instance. We also use $\mathcal{P}(y|x; f_\theta)$ to denote the probabilistic label distribution of $x$ given by $f_\theta$. Then AL selects the most informative sample batch and updates the labeled set accordingly. In the remainder of this paper, we omit parameter $f_\theta$ in $h_{acq}$ when the reliance on acquisitions over classifiers is clear.

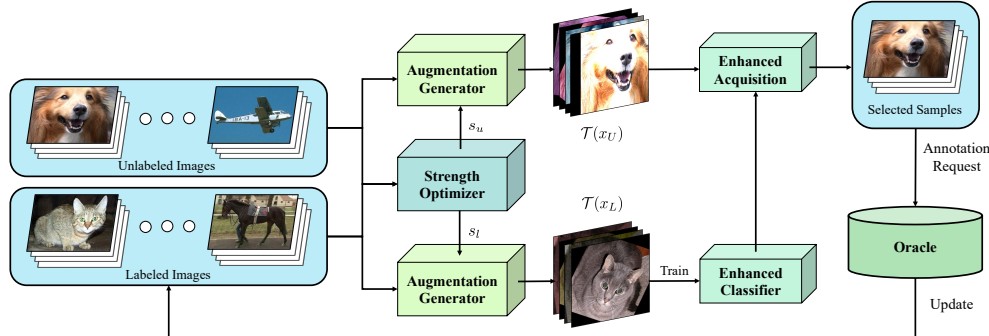

Figure 2: An active learning cycle for the CAMPAL framework. We optimize strengths $s_u, s_l$ for the unlabeled/labeled pool separately, then generate the corresponding strength-guided augmented views. We train an enhanced classifier over augmented labeled samples and induce an enhanced acquisition with augmented unlabeled samples. Finally, the acquisition selects informative samples to be labeled by an oracle.

**Data Augmentation.** We denote a single data augmentation (DA) operator by $T(x)$ that transforms a data point $x$ to another view via translation, rotation, or other augmentation operations. Then we denote the set consisting of these augmentations by $\mathcal{T}$. In practice, several studies also provide extended augmentation operators consisting of multiple operators (Hendrycks et al., 2019; Xu et al., 2022). The number of operators $s$ in the composition is named the *strength* of $T$, then $\mathcal{T}^{(s)}$ denotes the augmentation set with strength $s$. Intuitively, $s$ also quantifies how far an augmentation drifts images away from their original counterparts. Given a data point $x$, we also use $\mathcal{T}(x)$ to denote all its augmented views. With the notations given above, we continue our controllable augmentation-induced acquisition in the following sections.

## 2.2 CONTROLLABLE AUGMENTATIONS FOR ACTIVE LEARNING

As shown in Figure 1, data augmentation plays different roles in different data pools. On labeled data, it targets to improve the model prediction performance such that requires the generated virtual examples to be label invariant. In contrast, on unlabeled examples, it enlarges the exposed data distribution in the pursuit of better acquisition distribution. In this section, we propose a principled framework that searches for feasible DA configurations for different data pools in their own natural habits. It is worth noting that we adopt a dynamic control on the strength of augmentations across different cycles, making them adaptable to changes in AL as the cycle proceeds. It can be empirically verified that such dynamic control is better than a fixed augmentation strategy (Section 3.3). Through appropriate strength control, we expect to increase the quality of augmentations for AL.

**Strength for Unlabeled Data.** The primary goal for augmenting unlabeled data is to offer precise informativeness evaluation with an enriched distribution induced accordingly, thus inducing a more reliable acquisition. A problem with this is the invalid information introduced by potential undesirable augmentations. In detail, weak augmentations contain trivial augmentations that contribute little to the distribution enrichment, while drastic augmentations introduce excessive distribution drifts that mislead the acquisition. We resolve this problem by proposing a proper strength that maximizes the overall informativeness of the augmented unlabeled pool. Specifically, we maximize the least information augmentations can provide, ensuring that an optimized strength offers reliable informativeness, which can be formulated as follows:

$$s_u = \arg\max_s \sum\nolimits_{x_U \in \mathcal{D}_U} \min\{\mathbb{H}(\tilde{x}_U) \,|\, \tilde{x}_U \in \mathcal{T}^{(s)}(x_U), \, f_\theta(\tilde{x}_U) = f_\theta(x_U)\}, \qquad (1)$$

where $\mathbb{H}$ can be an arbitrary informativeness metric, and we adopt entropy here as it is sufficient to derive a proper $s_u$. By adopting a max-min optimization procedure, we can eliminate the potential negative impact brought by the corruption from aggressively augmented samples with $\min\{\mathbb{H}(\tilde{x}_U)\}$, and maximize the overall informativeness of the augmented unlabeled pool with $\arg\max$.

**Strength for Labeled Data.** By involving augmentations in model training, we aim at obtaining a dependable model from limited labeled data and further enhancing the acquisition process. Different from augmentation for unlabeled data that maximize overall informativeness, augmentations for labeled data are prone to training stability and convergence. To give out proper control over labeled

augmentations while avoiding extra training costs, we introduce a virtual loss term $\mathcal{L}_f$ and search the proper strength $s_l$ for labeled samples by minimizing it:

$$s_l = \arg\min_s \frac{1}{|\mathcal{D}_L|} \sum_{x_L \in \mathcal{D}_L} \mathcal{L}_f(x_L, s),$$

$$\text{where } \mathcal{L}_f(x, s) = \mathcal{L}(x) + \lambda_1 \,\mathrm{JS} \left( \{\mathcal{P}(y \,|\, \tilde{x}; f_\theta) \,|\, \tilde{x} \in \mathcal{T}_{single}^{(s)}(x)\} \right) + \frac{\lambda_2}{|\mathcal{T}_{mix}^{(s)}(x)|} \sum_{\dot{x} \in \mathcal{T}_{mix}^{(s)}(x)} \mathcal{L}(\dot{x}), \quad (2)$$

where $\mathcal{L}(x), \mathcal{L}_f(x)$ denotes the normal loss term and the augmented loss respectively, $\lambda_1, \lambda_2$ denotes fixed weights. For single-image augmentations $\mathcal{T}_{single}^{(s)}$, we integrate the augmented information into the model by making them produce similar outputs, in which the dissimilarity is quantified with a Jensen-Shannon (JS) divergence term. For Image Mixing $\mathcal{T}_{mix}^{(s)}$, we just follow the setup of Mixup.

With the strengths $s_u, s_l$ given above, we locate augmentations $\mathcal{T}_u$ that effectively enlarge the distribution, and the augmentations $\mathcal{T}_l$ that help deduce dependable classifiers. The combination of the two enables us to enhance acquisitions by making classifiers and informativeness evaluations in the AL framework work collaboratively and efficiently. We will show how augmentations for unlabeled samples (UA) and labeled samples (LA) contribute to the acquisition in Section 3.3.

## 2.3 CONTROLLABLE AUGMENTATION-INDUCED ACQUISITION FOR ACTIVE LEARNING

With the properly-controlled augmentations in Section 2.2, we proceed by providing a fast and efficient approach to integrate the functionalities of acquisitions and augmentations, i.e. controllable augmentation-induced acquisition. A key challenge for inducing the augmented acquisition $h_{acq}$ arises from the complicated forms for $h_{base}$, which denotes basic acquisitions and varies across different studies. In this section, we highlight two types of acquisitions, i.e., score-based acquisition and representation-based acquisition. We treat these two types of $h_{base}$ differently and describe the corresponding augmented acquisition forms. Notably, CAMPAL can adopt various kinds of acquisitions and enhance them, see Section 3. Since training a classifier $f_\theta$ with augmentations is straightforward, we focus on formulating augmented acquisition with augmented unlabeled data.

**Integrating Augmentations into Score-based Acquisitions.** Score-based acquisition calculates an information score for each data point and selects samples with the highest score, like Max Entropy (Settles, 2009). We enhance them by aggregating the information provided by augmentations, which are given by real value scores. Specifically, for methods that calculate an acquisition score $h_{base}(x)$ for each sample $x$, we calculate an information score $h_{base}(\tilde{x})$ for every augmented counterpart $\tilde{x} \in \mathcal{T}(x)$ and aggregate them into one score. We propose several variants of $h_{acq}$, including:

1. $h_{acq}(x) = \min_{\tilde{x} \in \mathcal{T}_u(x)} h_{base}(\tilde{x})$ reduces potential redundant information with a minimum acquisition score within the augmented batch;

2. $h_{acq}(x) = \sum_{\tilde{x} \in \mathcal{T}_u(x)} h_{base}(\tilde{x})$ sums up all the informativeness provided by augmentations;

3. $h_{acq}(x) = \sum_{\tilde{x} \sim \mathcal{T}_u(x)} \mathrm{sim}(x, \tilde{x}) h_{base}(x)$ weights the informativeness of $\tilde{x}$ by its similarity to its non-augmented counterpart, thus introducing the inter-sample information.

**Integrating Augmentations into Representation-based Acquisitions.** For representation-based acquisitions, $h_{base}$ provides a feature vector embedded into a representation space and performs sampling according to this space, like Core-set (Sener et al., 2018). Notice that representation-based methods rely on a distance function to measure the correlation between instances, we generalize the distance functions between individual samples to point-set distance functions between augmented sample batches. By adopting set distance functions, we enhance the acquisition process by taking the correlation across augmentations over different samples into consideration. To this end, we focus on well-defined set distance functions and propose the corresponding variants as follows:

1. Standard distance: $d(x, z) = \min_{\tilde{x} \in \mathcal{T}_u(x), \tilde{z} \in \mathcal{T}_u(z)} \|\tilde{x} - \tilde{z}\|_2^2$;

2. Chamfer distance: $d(x, z) = \sum_{\tilde{x} \in \mathcal{T}_u(x)} \min_{\tilde{z} \in \mathcal{T}_u(z)} \|\tilde{x} - \tilde{z}\|_2^2 + \sum_{\tilde{z} \in \mathcal{T}_u(z)} \min_{\tilde{x} \in \mathcal{T}_u(x)} \|\tilde{x} - \tilde{z}\|_2^2$ considers pairwise similarities for the augmented views from two samples;

3. Pompeiu–Hausdorff distance: $d(x, z) = \max\{\max_{\tilde{x} \in \mathcal{T}_u(x)} d(\tilde{x}, \mathcal{T}_u(z)), \max_{\tilde{z} \in \mathcal{T}_u(z)} d(\mathcal{T}_u(x), \tilde{z})\}$ highlights the maximal potential difference between two samples.

Table 1: Comparison of the averaged test accuracy on benchmark datasets and different AL strategies. Since CAMPAL has multiple versions, we choose the one with the best performance and denote it with CAMPAL*. The best performance in each category is indicated in boldface. $N_L$ denotes the number of labeled samples.

| Dataset | Method | $N_L = 500$ | $N_L = 1,000$ | $N_L = 1,500$ | $N_L = 2,000$ |
|---------|--------|-------------|---------------|---------------|---------------|
| SVHN | Random | 52.42±2.10 | 64.38±1.91 | 68.55±1.30 | 71.43±1.34 |
| | Entropy | 55.86±1.66 | 66.42±2.49 | 73.08±2.84 | 75.40±2.43 |
| | BADGE | 56.19±1.97 | 67.30±2.19 | 76.35±0.57 | 80.03±1.68 |
| | BGADL | 40.18±0.43 | 50.58±1.30 | 64.56±1.34 | 69.73±1.34 |
| | CAL | 56.98±1.07 | 66.22±0.92 | 72.09±1.83 | 75.22±2.11 |
| | LADA | 56.61±1.50 | 66.56±1.21 | 72.48±1.66 | 75.84±1.12 |
| | CAMPAL* | **61.34±4.26** | **78.81±0.93** | **82.86±0.42** | **85.66±0.79** |
| CIFAR-10 | Random | 38.54±2.28 | 49.77±3.08 | 58.61±2.75 | 61.49±2.06 |
| | Entropy | 39.80±1.60 | 55.43±1.71 | 60.76±2.64 | 65.95±1.36 |
| | BADGE | 44.18±2.09 | 55.97±1.57 | 62.40±2.15 | 67.03±0.62 |
| | BGADL | 37.54±1.88 | 47.57±1.38 | 51.81±1.00 | 56.73±0.75 |
| | CAL | 40.05±1.68 | 54.24±2.30 | 59.83±2.66 | 64.24±0.91 |
| | LADA | 41.87±2.33 | 56.37±2.24 | 62.76±1.99 | 66.26±1.29 |
| | CAMPAL* | **52.26±2.01** | **73.36±1.11** | **77.87±0.61** | **80.37±0.86** |

(a) SVHN     (b) CIFAR-10     (c) CIFAR-100

Figure 3: Test accuracy on the number of labeled samples over different datasets.

**Controllable DA-Driven Active Learning Cycles.** With those augmentation-induced acquisitions, we complete the active learning cycle within CAMPAL. First, we generate the labeled augmentations $\mathcal{T}_l$ with properly controlled strength $s_l$, then produce an augmented classifier $f_\theta$ trained over them. This makes up for the insufficient labeled information and further brings a reliable model. Second, we generate the unlabeled augmentations with an optimized strength $s_u$ and induce the enhanced acquisition $h_{acq}$ with $\mathcal{T}_u$ and $f_\theta$. Notably, CAMPAL offers a dynamic strength control on augmentations across cycles, which also leads to a controllable acquisition adapting itself to the changing data pools. This augmentation-induced acquisition step provides precise information evaluation and guarantees the positive impact of augmentations, which finally helps produce better querying results. As a result, these two steps jointly ensure the quality of data to label at the end of the active learning cycle, largely boosting the performance. Our experiments in Section 3 show their separate effects as well as the combined impacts in detail. The pseudo-code of our algorithm is provided in Appendix C.

# 3 EXPERIMENTS

## 3.1 BASELINES AND DATASETS

We instantiated our proposed CAMPAL with several existing strategies, including 1) Entropy, 2) Least Confidence (LC), 3) Margin, 4) Core-set (Sener et al., 2018), and 5) BADGE (Ash et al., 2020). We also implement several augmentation-aggregation modes that integrate augmentations into an enhanced acquisition, including 1) MIN, 2) SUM, 3) DENSITY for Entropy, LC, Margin, and 1) STANDARD, 2) CHAMFER, 3) HAUSDORFF for Core-set, BADGE, as shown in Section 2.3 and Table 2. In this section, we specify the instantiated augmentation-acquisition with basic strategy $h_{base}$ as its subscript and the augmentation-aggregation mode as its superscript, e.g. $\text{CAMPAL}_{\text{Entropy}}^{\text{MIN}}$.

Table 2: Performance of CAMPAL with different $h_{base}$ and aggregation modes. The experiment is conducted over CIFAR-10 with 2,000 labeled samples.

| Method | | Aggregation Mode | | |
|---|---|---|---|---|
| Type of $h_{base}$ | $h_{base}$ | MIN | SUM | DENSITY |
| Score | Entropy | 76.90±0.76 | 75.87±0.32 | 78.89±0.74 |
| | LC | 76.82±0.62 | 72.74±1.31 | 76.76±0.54 |
| | Margin | 78.70±0.58 | 71.33±0.76 | 79.16±0.48 |
| Type of $h_{base}$ | $h_{base}$ | STANDARD | CHAMFER | HAUSDORFF |
| Representation | Core-set | 78.20±0.28 | 79.67±0.68 | 78.49±0.51 |
| | BADGE | 79.71±0.51 | 80.37±0.86 | 79.84±0.24 |

Table 3: Comparison of CAMPAL with its non-augmented counterpart with different AL strategies. $\Delta$ indicates the performance boost brought by CAMPAL.

| Dataset | Method | Entropy | LC | Margin | Coreset | BADGE |
|---|---|---|---|---|---|---|
| Fashion | baseline | 81.33±0.86 | 81.15±1.16 | 80.71±1.16 | 83.36±0.82 | 82.89±0.95 |
| | +CAMPAL | 85.89±0.29 | 84.63±1.31 | 84.82±0.62 | 84.36±0.48 | 86.24±1.04 |
| | $\Delta$ | +4.56±0.91 | +3.48±1.75 | +4.11±1.32 | +1.00±0.95 | +3.35±1.41 |
| SVHN | baseline | 75.40±2.43 | 76.39±1.30 | 76.32±1.87 | 77.81±0.93 | 80.03±1.68 |
| | +CAMPAL | 85.36±0.45 | 84.34±0.62 | 84.90±0.57 | 84.35±0.81 | 85.66±0.79 |
| | $\Delta$ | +9.96±2.47 | +7.95±1.44 | +8.58±1.95 | +6.54±1.23 | +5.63±1.86 |
| CIFAR-10 | baseline | 65.95±1.36 | 66.97±1.87 | 66.76±1.77 | 66.90±0.93 | 67.03±0.62 |
| | +CAMPAL | 78.89±0.74 | 76.82±0.62 | 79.16±0.48 | 79.67±0.68 | 80.37±0.86 |
| | $\Delta$ | +12.94±1.55 | +9.85±1.97 | +12.40±1.83 | +12.77±1.15 | +13.34±1.06 |
| CIFAR-100 | baseline | 45.18±0.13 | 45.70±0.18 | 45.64±0.25 | 46.52±0.21 | 47.75±0.09 |
| | +CAMPAL | 48.76±0.30 | 49.24±0.70 | 49.63±0.65 | 46.80±0.27 | 49.46±0.65 |
| | $\Delta$ | +3.58±0.33 | +3.54±0.72 | +3.99±0.70 | +0.28±0.34 | +1.71±0.66 |

We also denote the version with the best performance for CAMPAL, i.e. $\text{CAMPAL}_{\text{BADGE}}^{\text{CHAMFER}}$ as CAMPAL*, as shown in Table 2. We repeat every experiment 5 times.

In this work, we compare our method to 1) Random, 2) Coreset, 3) BADGE, 4) Max Entropy, 5) Least Confidence 6) Margin. We also compare our method with other active learning strategies with data augmentations, including 1) BGADL (Tran et al., 2019), 2) CAL (Gao et al., 2020), and 3) LADA (Kim et al., 2021b) in Table 1. For a fair comparison, CAL does not use its original semi-supervised setting but uses a supervised procedure. Since LADA has multiple versions, we choose the one with the best performance for comparison in Table 1. We further prove the efficacy of CAMPAL by comparing its performance with the corresponding baseline versions in Table 3.

## 3.2 MAIN EMPIRICAL RESULTS

**CAMPAL achieves SOTA results.** As shown in Table 1 and Figure 3, CAMPAL significantly outperforms their rivals on many datasets and data scales. Specifically, on the CIFAR-10 dataset, we improve upon the best baseline by **8.08**%, **16.99**%, **15.11**%, **13.34**%, where the labeled set has 500, 1000, 1500, 2000 instances respectively. Moreover, CAMPAL exhibits the most significant performance boost with a moderately small $N_{\mathcal{L}}$, which is approximately around 1,000 for CIFAR-10 and SVHN. Besides, we can see that different versions of CAMPAL consistently achieve superior results on CIFAR-10, as shown in Table 2. As shown in Table 3, in all combinations of baselines and datasets, CAMPAL variations exhibit the best performance. Notably, CAMPAL also brings a consistent performance boost with strong scalability.

In addition, it is worth noting that previous works (Tran et al., 2019; Kim et al., 2021a) are typically evaluated with a large number of labeled samples (e.g., $10\% \sim 40\%$ of labeled samples for CIFAR-10). We also challenge this by querying fewer samples over benchmark datasets, shown in Table 1. When $N_{\mathcal{L}} = 500$ or 2,000 on CIFAR-10, recent augmentation-based AL strategies fail to outperform other simple baselines like BADGE. Notably, BGADL performs the worst, because of the inadequate training with insufficient instances in the current active learning setting. Since CAL is originally designed for a semi-supervised setting, it fails to outperform simple baselines like BADGE under

Table 4: Test accuracy of CAMPAL when UA or LA are individually applied over CIFAR-10 with 2,000 labeled samples. The results are produced over 5 different AL strategies.

| Components | | Entropy | LC | Margin | Core-set | BADGE |
|---|---|---|---|---|---|---|
| UA | LA | | | | | |
| | | 65.95±1.36 | 66.97±1.87 | 66.76±1.77 | 66.90±0.93 | 67.03±0.62 |
| ✓ | | 67.49±1.87 | 69.59±2.55 | 71.86±3.16 | 68.83±1.29 | 71.24±0.75 |
| | ✓ | 74.30±0.94 | 75.92±0.85 | 77.73±0.44 | 77.73±0.20 | 78.89±0.22 |
| ✓ | ✓ | **76.90±0.76** | **76.82±0.62** | **78.70±0.58** | **78.20±0.28** | **79.71±0.51** |

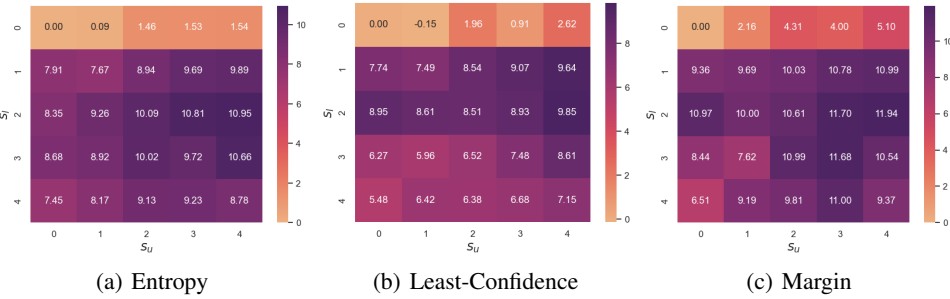

(a) Entropy      (b) Least-Confidence      (c) Margin

Figure 5: A heatmap visualization of performance boost brought by augmentations of different strengths, when attached to the labeled pool and unlabeled pool. The experiments are performed over CIFAR-10 with 2,000 labeled samples and are conducted over Entropy, LC, and Margin.

our supervised setting. LADA outperforms other baselines on CIFAR-10 but fails on SVHN since maximal-entropy augmentations can easily change the semantics of the digit data. In contrast, our proposed CAMPAL remains competitive, indicating its superiority.

**The learnt augmentation strength differs for unlabeled/labeled data.** In Figure 4, we visualize the dynamics of the learnt strength $s_u^*, s_l^*$ across active learning cycles. In particular, we conduct the experiment 5 times on CIFAR-10 with $\text{CAMPAL}_{\text{Entropy}}^{\text{MIN}}$ and $\text{CAMPAL}_{\text{BADGE}}^{\text{STANDATD}}$ and figure out the average optimal strength value. We can observe that the $s_u^*$ is generally larger in comparison with $s_l^*$ across the AL cycles. This verifies our postulations that labeled requires moderate augmentation for label preserving. In contrast, unlabeled data prefers relatively stronger augmentations to enrich the data distribution such that a wider range of informative regions can be explored. We conclude that AL is better enhanced by

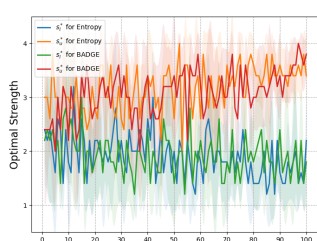

Figure 4: The learned strength on CIFAR-10, with CAMPAL instantiated with Entropy/BADGE.

DA with a combinatorial scheme of weak and strong augmentations applied to labeled and unlabeled data in our framework, which also corroborates our theoretical findings in Section 4.

### 3.3 EMPIRICAL ANALYSIS

In this section, we present our ablation results to show the effectiveness of our framework. We exemplify the superiority of your CAMPAL by using scored-based acquisition with MIN as the aggregation; see Appendix B for more ablation experiments.

**Impact of Unlabeled/Labeled Augmentations.** Here, we compare the performance boost of augmentation-induced acquisitions based on different AL strategies and the results are reported in Table 4. We can see that without augmented labeled information, the enhanced acquisition gives out a consistent performance boost over several strategies, and the maximal boost is presented by Margin ($\Delta 5.10\%$). The enhanced training process also plays an important role in promoting the performance of the existing strategies by $8.35\% \sim 11.86\%$. A combination of these two components also shows consistently best performance compared to other ablation versions, indicating that they can work well with each other and unleashes different types of information. We can conclude that both the augmented unlabeled information and the labeled ones help resolve the problem of unreliable judgment in AL strategies.

**Compare with fixed augmentation strengths.** Since we emphasize the importance of a strength control over $s_l, s_u$ in Section 2.2, we will provide more details here. In brief, augmentations with various strengths contribute to the performance but can be inefficient when strengths are not chosen appropriately. To further look at the impact of augmentation sets with different strengths, we fix the value for $s_l, s_u$ and see how they decide the final performance. Specifically, we test different combinations of $s_l$ and $s_u$ in the range $[0, 4]$, with other settings following the main empirical studies. The relative performance boost compared to their non-augmented counterparts is shown in Figure 5. Without proper strength control, the performance boost can decrease. For instance, $\text{CAMPAL}_{\text{Margin}}^{\text{MIN}}$ with $s_l = 3, s_u = 1$ leads to a 4.32% performance drop compared to the optimal one, when the worst case in $\text{CAMPAL}_{\text{Entropy}}^{\text{MIN}}$ causes a 3.28% drop similarly. In addition, we can also see a trend similar to Section 3.2 that the classifier $f_\theta$ prefers weakly labeled augmentations when stronger unlabeled augmentations induce stronger acquisitions, even without a dynamic strength control.

# 4 THEORETICAL ANALYSIS

In this section, we theoretically analyze why weak and strong augmentations being strategically applied to labeled and unlabeled data exhibit the best performance when combining AL with DA. Following the previous sections, we use $f_\theta$ to denote the model fully trained over augmented labeled samples. When an unlabeled sample lies within the augmented region for a particular labeled sample, we can propagate the labeled information to the corresponding unlabeled samples. Formally, with a feature map $f_\theta^{\text{emb}}$ derived from $f_\theta$ we define a covering relation between augmented labeled batches and unlabeled samples as follows:

**Definition 1.** *Given a collection of augmentations $\mathcal{T}$, we say that an image $x$ is covered by $x_i$ with respect to the augmentation set $\mathcal{T}$, if the feature embedding of $x$ lies within the convex hull of the augmented views of $x_i$: $f_\theta^{\text{emb}}(x) \in conv\left(f_\theta^{\text{emb}}\left(\mathcal{T}(x_i)\right)\right)$. We denote the covering relation by $x \lhd x_i$.*

Without loss of generality, assume there are $L$ labeled samples $x_1, \ldots, x_L$, together with the unlabeled samples covered by its augmentations, constituting $L$ components. For each component $C_i(i = 1, \ldots, L)$, let $P_i$ be a probability that a data point sampled from the underlying data distribution covered by $C_i$. To make the analysis tractable, we assume the properly controlled augmentations for labeled samples, eliminating the potential overlaps across different components:

**Assumption 2.** *With moderately weak augmentations for labeled samples, $C_i$'s do not overlap with each other; i.e. $\forall i \neq j, \mathcal{P}(C_i \cap C_j \neq \emptyset) = 0$.*

With Assumption 2, the error for $f_\theta$ can be estimated by how these components cover the data space. To further illustrate this, we provide a comparison between different augmentations in Figure 6. The following proposition characterizes the relationship between the error and the components.

**Proposition 3.** *Let $\mathcal{E}$ denote the probability that the $f_\theta$ cannot infer the correct label of a test example. Then $\mathcal{E}$ is upper bounded by*

$$\mathcal{E} \leq \sum_{i=1}^{L} P_i(1 - P_i)^m + \left(1 - \sum_{i=1}^{L} P_i\right), \tag{3}$$

*where $m$ denotes the number of samples that lie within the labeled components.*

In Eq. (3), the first term denotes the risk brought by ill-defined augmentations, while the second term denotes sub-sample empirical risk. With Eq. (3), we continue to reduce the error as much as possible by acquiring informative samples. By adding a newly queried sample $x_{L+1}$, the error reduction is estimated as follows:

$$\Delta\mathcal{E}(\Delta m, P_{L+1}) \approx \sum_{i=1}^{L} P_i(1 - P_i)^m \left(1 - (1 - P_i)^{\Delta m}\right) - P_{L+1}\left(1 - (1 - P_{L+1})^{m+\Delta m}\right), \tag{4}$$

where $\Delta m$ is the number of samples newly covered after labeling $x_{L+1}$.

We take a step further by illustrating two terms in Eq. 4. The first term denotes the performance boost brought by better coverage with newly-annotated samples. Specifically, the samples that drift farthest from the existing components better cover the under-explored data space, indicating a larger $\Delta m$ – in turn – the performance boost. This is also consistent with the max-min optimization objective for unlabeled samples described in Eq. (1), with the intuition provided in Figure 6(c),(d). The second

term characterizes the potential error induced from augmentations on unlabeled samples, i.e., too strong augmentation excessively increases the value of $P_{L+1}$, leading to its increase. Therefore, it is important to locate moderately strong augmentations for unlabeled data in AL.

**Theorem 4.** *With properly selected augmentation sets and sufficient large $L$, the maximal value for error reduction $\Delta\mathcal{E}(\Delta m, P_{L+1})$ with newly-annotated samples can be estimated as follows:*

$$\Delta\mathcal{E}(\Delta m, P_{L+1}) \lesssim \mathcal{E}\left(1 - Ke^{-m/L}\right), \tag{5}$$

*where $U$ denotes the number of unlabeled samples, with $K = \frac{m + L(\log(L+U) - \log L - 1)}{L+U}$.*

From the theorem, we can see that properly selected samples and augmentations give out a significant error reduction. Specifically, $m/L$ denotes the average number of samples covered by each component, which indicates better coverage induced from properly controlled components when being larger. Revisiting our theoretical proof, we further explain that DA indeed serves different goals in AL. On the one hand, the augmentations on labeled data guarantee that Assumption 2 holds, indicating that we need a dependable model and weak augmentations. On the other hand, this theorem emphasizes the importance of acquiring newly-labeled samples guided by moderately strong augmentation, ensuring better coverage while also avoiding potential misleading information. With all the discussions above, augmentation-acquisition integration effectively relies on the quality of augmentations, where better augmentations result in more dependable classifiers for AL and larger error reduction across AL cycles. This echoes our discussion of the benefit of appropriately controlled data augmentations for AL. A more detailed analysis is given in Appendix A.

## 5 RELATED WORKS

**Data Augmentation** is a technique that improves the generalization ability of models by increasing the number of images and their variants in a dataset (Xu et al., 2022). The most commonly used augmentation techniques include geometric transformations (Shorten & Khoshgoftaar, 2019), random erasing (Devries & Taylor, 2017; Zhong et al., 2020) and generative adversarial networks (Zhu et al., 2018; Bowles et al., 2018). Another type of augmentation is image mixing (Zhang et al., 2018; Yun et al., 2019), which blends multiple images and their corresponding labels. Instead of designing new types of augmentations, recent studies also collect a group of augmentations and optimize their strength (a.k.a strength) (Cubuk et al., 2020), which quantifies how far an augmented image drift from its original counterpart. By optimizing this strength, several studies attain state-of-the-art performance over several benchmarks (Zheng et al., 2022; Yang et al., 2022).

**Active Learning** is a machine learning paradigm in which a learning algorithm actively selects the data it wants to learn from the unlabeled data sources (Settles, 2009; Ren et al., 2021). The crucial part of active learning in most existing strategies is exactly the data acquisition process, which targets selecting the most informative examples. Current studies mostly focus on specific parts of samples and can be roughly categorized as follows: (a) Uncertainty-based methods that prefer the hardest samples (Choi et al., 2021; Mai et al., 2022) or the ones the current fully-trained model uncertain about (Kirsch et al., 2019; Wang et al., 2022); (b) Representation-based methods searching for the samples that are the most representative of the underlying data distribution (Sener et al., 2018; Ash et al., 2020; Kim & Shin, 2022). To date, the unreliable informativeness evaluation with very few samples remains a critical issue for Active Learning.

## 6 CONCLUSIONS

In this work, we propose a novel active learning framework CAMPAL. Based on the observation that the impacts of augmentations applied to the disparate data pools differ due to their different goals, CAMPAL conducts appropriate controls on data augmentation integrated into active learning. We empirically find CAMPAL attains state-of-the-art performance with a significant performance boost, especially with fewer labeled samples. Our theoretical analysis further guarantees this difference and claims the reliance of AL on the quality of introduced augmentations. In the future, we hope to generalize CAMPAL to more tasks and investigate the impact of DA over AL in more detail.

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

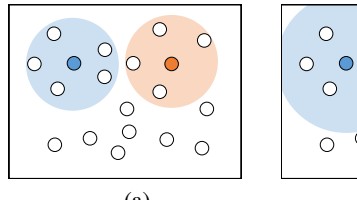 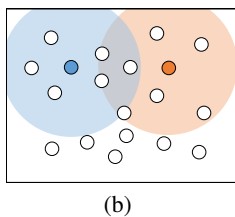 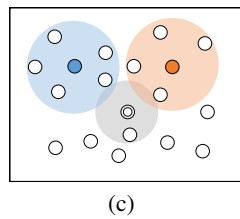 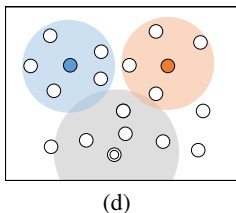

(a)  (b)  (c)  (d)

Figure 6: The coverage on the data space presented by augmentations, where colored circles are labeled samples, white circles are unlabeled samples and the colored shade denotes the region covered by corresponding augmentations. A double circle denotes the unlabeled sample to be annotated. The figures above show: a) Proper augmentation for labeled samples; b) Drastic augmentation for labeled samples; c) Sub-optimal unlabeled sample with the corresponding augmentation; d) A proper unlabeled sample with the corresponding augmentation.

# A   THEORETICAL ANALYSIS

This section provides a complete derivation for the analysis given in Section 4. An intuition for this is given in Figure 6. Before the actual acquisition process, we must ensure convergence for the underlying classifier. Specifically, with proper augmentations over labeled data and the approximate loss term in equation 3, we can deduce the upper bound for $Pr(\mathcal{A})$ and guarantee the convergence for training, shown as follows:

**Theorem 1.** *Under the setting for CAMPAL, Let $\mathcal{E}$ denote the probability that the classifier cannot infer the label of newly given samples drawn from the underlying data space, with L labeled samples given in $\mathcal{D}_L$ and augmentation set $\mathcal{T}$. Then $\mathcal{E}$ is upper bounded by $\hat{\mathcal{E}}$ as follows:*

$$\mathcal{E} \leq \hat{\mathcal{E}}(\mathcal{D}_L, f_\theta, \mathcal{T}) = \sum_{l=1}^{L} P_i(1-P_i)^m + \left(1 - \sum_{i=1}^{L} P_i\right), \tag{6}$$

*With properly selected augmentation set $\mathcal{T}$ and sufficient large L, $\hat{\mathcal{E}}$ can be estimated by $O(\varepsilon)$ with $O(L/\varepsilon)$ samples covered by labeled components, i.e.*

$$m = O(L/\varepsilon) \Rightarrow \hat{\mathcal{E}} \lessapprox O(\varepsilon) \Rightarrow \mathcal{E} \leq O(\varepsilon). \tag{7}$$

*Proof.* With proper control over augmentations, we assume that each component does not overlaps with at most one other component in Proposition 3, which can be controlled with appropriate augmentations, and generalizable to multiple components. Let $x$ be the sampled example, the probability of $x$ not covered only in one of $C_i$'s is

$$\hat{\mathcal{E}} = \mathcal{P}\left(\exists i \neq j, x' \in C_i \cap C_j\right) + \mathcal{P}\left(x' \text{ is uncovered}\right)$$

$$= \sum_{l=1}^{L} P_i(1-P_i)^m + \left(1 - \sum_{i=1}^{L} P_i\right)$$

With sufficiently large L, we can also have a component set that covers the entire dataset, leading to $\sum_i P_i = 1$. Now it remains to find the maximum value of $\sum_{l=1}^{L} P_i(1-P_i)^m$ to bound the error term, with the following optimization objective:

$$\min_{C} - \sum_i P_i(1-P_i)^m, \text{ s.t. } \sum_i P_i = 1.$$

With the KKT condition, we attain its maximum value when all $P_i$ is set to $\frac{1}{L}$, i.e. $\hat{\mathcal{E}} \lessapprox (1 - \frac{1}{L})^m$. With $O(L/\varepsilon)$ and sufficiently large L, we have

$$\hat{\mathcal{E}} \lessapprox \exp\left(-\frac{m}{L}\right) = \exp\left(-O(\frac{1}{\varepsilon})\right) \leq O(\varepsilon).$$

$\square$

With the conditions in theorem 1, it remains to consider the approximate boost provided by the reduction on upper bound $\hat{\mathcal{E}}$:

$$\Delta\hat{\mathcal{E}}(\Delta m, P_{L+1}) = \hat{\mathcal{E}}(\mathcal{D}_L \cup \{x_{L+1}\}, f_\theta, \mathcal{T}\}) - \hat{\mathcal{E}}(\mathcal{D}_L, f_\theta, \mathcal{T}))$$

$$= \sum_{i=1}^{L} P_i(1-P_i)^m \left(1 - (1-P_i)^{\Delta m}\right) - P_{L+1}\left(1 - (1-P_{L+1})^{m+\Delta m}\right).$$

where $\Delta m$ is the number of samples newly covered after labeling $x_{L+1}$.

**Theorem 2.** *With the conditions given in Theorem 1, the maximal value for error bound reduction $\Delta\hat{\mathcal{E}}(\Delta m, P_{L+1})$ with newly-annotated samples can be estimated as follows:*

$$\Delta\mathcal{E}(\Delta m, P_{L+1}) \lessapprox \mathcal{E}\left(1 - Ke^{-m/L}\right), \tag{8}$$

*where $U$ denotes the number of unlabeled samples, with*

$$K = \frac{m\log(L+U) - L\left(\log L - 1\right)}{L+U}$$

.

*Proof.* Under this setting, $P_{L+1}$ appears to be proportional to $\Delta m$, when no unnecessary overlap appears across components (guaranteed by Theorem 1). Therefore, we can estimate $P_{L+1} \approx \Delta m/(L+U)$, where $U$ denotes the number of unlabeled samples. With those conditions, we estimate the relative error reduction as follows:

$$\frac{\Delta\hat{\mathcal{E}}}{\hat{\mathcal{E}}} = \frac{\sum_{i=1}^{L} P_i(1-P_i)^m \left(1 - (1-P_i)^{\Delta m}\right) - P_{L+1}\left(1 - (1-P_{L+1})^{m+\Delta m}\right)}{\sum_{i=1}^{L} P_i(1-P_i)^m + \left(1 - \sum_{i=1}^{L} P_i\right)}$$

$$\lessapprox \left(1 - \left(1 - \frac{1}{L}\right)^{\Delta m}\right) - \exp\left(-\frac{m}{L}\right)\frac{\Delta m}{L+U}\left(1 - \left(1 - \frac{\Delta m}{L+U}\right)^{m+\Delta m}\right)$$

Since $m$ is large with sufficient labeled samples, we can further estimate this term as:

$$\frac{\Delta\hat{\mathcal{E}}}{\hat{\mathcal{E}}} \lessapprox 1 - \left(1 - \frac{1}{L}\right)^{\Delta m} - \exp(-\frac{m}{L})\frac{\Delta m}{L+U}.$$

Then the maximum value for this is attained when $\Delta m$ reaches

$$\Delta m^* = \frac{1}{\log\left(1 - \frac{1}{L}\right)}\left(\frac{m}{L} - \log\left(\frac{1}{L+U}\right) - \log\left(-\log\left(1 - \frac{1}{L}\right)\right)\right)$$

$$\approx L\left(\frac{m}{L} - \log\left(\frac{1}{L+U}\right) + \log(\frac{1}{L})\right) = m + L\left(\log\left(L+U\right) - \log L\right)$$

Then

$$\frac{\Delta\hat{\mathcal{E}}}{\hat{\mathcal{E}}} \approx 1 + \frac{1}{L+U}\left(\frac{1}{\log(1 - \frac{1}{L})} - \Delta m^*\right)\exp\left(-\frac{m}{L}\right)$$

$$\approx 1 - \frac{m + L\left(\log(L+U) - \log L - 1\right)}{L+U}\exp\left(-\frac{m}{L}\right).$$

$\square$

# B  ADDITIONAL EXPERIMENTAL SETUPS AND RESULTS

## B.1  IMPLEMENTATION DETAILS

We conduct experiments on four benchmark datasets: FashionMNIST, SVHN, CIFAR-10, and CIFAR-100. We will construct a random initial dataset with 100 instances for FashionMNIST,

Table 5: The list of all the augmentations used in the experiments. The letter $x$ or $x_*$ denotes given images. $\mathcal{U}(a, b)$ denotes a continuous uniform distribution at interval $[a, b]$, when $\mathcal{B}(a, b)$ denotes a beta distribution with parameters $a$ and $b$.

| Augmentation | Parameters | Description |
|---|---|---|
| AutoContrast($x$) | | Maximizing the (normalize) image contrast |
| Brightness($x,v$) | $v \sim \mathcal{U}(1, 1.18)$: an enhancing factor | Enhancing the brightness of a given image |
| Color($x,v$) | $v \sim \mathcal{U}(1, 1.18)$: an adjustment factor | Adjust the color balance of a given image |
| Contrast($x,v$) | $v \sim \mathcal{U}(1, 1.18)$: | Enhancing the contrast of a given image |
| CutOut($x,v$) | $v \sim \mathcal{U}(0.09, 0.11)$: CutOut ratio | Cut out a part of image and fill with black |
| CutOutAbs($x,v$) | $v \sim \mathcal{U}(0.09, 0.11)$: CutOut ratio | Cut out a part of image and fill with gray |
| Equalize($x$) | | Equalize the image histogram |
| Identity($x$) | | Return the image itself |
| Invert($x$) | | Invert all pixel values |
| Posterize($x,v$) | $v \sim \mathcal{U}(6.0, 6.4)$: Posterizing degree | Posterizing the image |
| Rotate($x,v$) | $v \sim \mathcal{U}(20, 30)$: Rotation degree | Rotating the image |
| Sharpness($x,v$) | $v \sim \mathcal{U}(1, 1.18)$: Sharpen degree | Sharpen the image |
| ShearX($x,v$) | $v \sim \mathcal{U}(0.15, 0.18)$: Affinity degree | Affine transformation in x-axis |
| ShearY($x,v$) | $v \sim \mathcal{U}(0.15, 0.18)$: Affinity degree | Affine transformation in y-axis |
| Solarize($x,v$) | $v \sim \mathcal{U}(96, 128)$: Solarization degree | Solarizing the image |
| SolarizeAdd($x,v$) | $v \sim \mathcal{U}(50, 60)$: Solarization degree | Solarizing the image and add back |
| TranslateX($x,v$) | $v \sim \mathcal{U}(0.1, 0.15)$: translation ratio | Translating the image in x-axis |
| TranslateY($x,v$) | $v \sim \mathcal{U}(0.1, 0.15)$: translation ratio | Translating the image in y-axis |
| MixUp($x,x_*, \lambda$) | $\lambda$: the mixing ratio | Mix up the two given images |

SVHN, and CIFAR-10, and 1,000 instances for CIFAR-100. Then we acquire 100 instances for FashionMNIST, SVHN, and CIFAR-10, and 500 instances for CIFAR-100 at each cycle. We repeat the cycle 20 times. Then we generate 10 single-image augmentations and 5 mix-up augmentations for each sample. We normalize the images with the channel mean and standard deviation over all the datasets. For CIFAR-10 and CIFAR-100, we apply a standard augmentation after conducting augmentations in the pipeline. We adopt ResNet-18 as the architecture and train the model for 300 epochs with an SGD optimizer of learning rate 0.01, momentum 0.9, and weight decay 5e-4. For the virtual loss term in equation 2, we also set $\lambda_1 = \lambda_2 = 1$.

## B.2 Implementation Details for the simple application of DA for AL in Figure 1

We integrate DA into AL with fixed augmentations $\mathcal{T}$ as follows. This experiment is also conducted on dataset CIFAR-10 with a ResNet-18 architecture. The basic acquisition here is Max Entropy. First, we augment the labeled pool with $\mathcal{T}$, and train the classifier $f_\theta$ accordingly. Then we augment the unlabeled pool with $\mathcal{T}$ and performs acquisitions directly on the augmented unlabeled pool. Other settings are the same as the main empirical experiments.

## B.3 Augmentations Included

The details of the 19 augmentations in the (CAMPAL) with their parameters are shown in Table 5. In brief, the augmentations we use can be categorized into single-image augmentations and image-mixing. Formally, we provide an augmentation functional set that covers (i)-singular input augmentation means such as rotation for low-level image processing. The corresponding functional set is denoted by $\mathcal{T}_{single} = \{\omega(x; \lambda)\}$ where $\omega$ points to an instantiated augmentation function. The sample $x$ is taken as an input to $\omega$ together with varying augmentation hyper-parameters $\lambda$, such as the angle in the image rotation function. Similarly, we also construct a combinatorial augmentation functional set, $\mathcal{T}_{mix} = \{\gamma(x, x'; \lambda)\}$, where the augmentation function $\gamma$ takes two input samples $x$ and $x'$ together with hyper-parameter. With slight abuse of notations, we uniformly use $\lambda$ to refer to augmentation-related hyper-parameters. In the implementation of CAMPAL, we simply adopt MixUp for combinatorial augmentation. As we can see, upon fixed input, both singular and combinatorial augmentation functional sets can be arbitrarily expanded, by varying $\lambda$ in a continuous scalar space.

Table 6: Test accuracy of CAMPAL and augmentation-induced acquisition with learned RandAugment.

| Method | Fashion | SVHN | CIFAR-10 | CIFAR-100 |
|---|---|---|---|---|
| Ent w. RA | 86.15±0.89 | 82.84±1.12 | 76.83±0.82 | 46.70±0.34 |
| CAMPAL$_{\text{Entropy}}^{\text{DENSITY}}$ | 86.17±0.58 | 83.49±0.96 | 78.89±0.74 | 48.76±0.30 |

Table 7: Comparison of the averaged test accuracy when each type of augmentation is separately integrated into CAMPAL. We ran each experiment on CIFAR-10 with 2,000 samples annotated at the last cycle, and repeat them 5 times. $\mathcal{N}_L$ denotes the number of labeled samples.

| Augmentation | $\mathcal{N}_L = 500$ | $\mathcal{N}_L = 1,000$ | $\mathcal{N}_L = 1,500$ | $\mathcal{N}_L = 2,000$ |
|---|---|---|---|---|
| None | 39.80±1.60 | 55.43±1.71 | 60.76±2.64 | 65.95±1.36 |
| AutoContrast | 48.59±1.12 | 63.35±0.30 | 72.77±0.12 | 76.36±0.28 |
| Brightness | 45.63±0.26 | 60.16±2.25 | 69.50±0.26 | 74.25±0.16 |
| Color | 50.84±3.50 | 62.04±1.75 | 72.24±0.25 | 76.64±0.46 |
| Contrast | 49.77±3.91 | 56.52±1.32 | 68.95±1.61 | 74.47±0.63 |
| CutOut | 47.91±2.66 | 62.29±3.69 | 71.37±1.19 | 76.62±0.34 |
| CutOutAbs | 53.59±0.30 | 63.96±0.57 | 67.94±0.19 | 71.94±0.18 |
| Equalize | 49.38±1.93 | 63.18±1.86 | 69.86±0.78 | 74.25±0.27 |
| Invert | 51.73±0.29 | 63.65±0.12 | 71.89±0.33 | 75.98±0.28 |
| Posterize | 49.02±1.16 | 64.24±1.30 | 72.25±0.90 | 75.62±0.64 |
| Rotate | 44.60±1.39 | 56.47±0.21 | 63.08±1.22 | 67.60±0.45 |
| Sharpness | 47.31±2.35 | 62.40±1.15 | 70.98±1.26 | 74.38±2.46 |
| ShearX | 45.19±0.39 | 58.22±0.99 | 67.96±0.54 | 72.16±1.20 |
| ShearY | 48.09±4.96 | 62.13±1.98 | 70.98±0.05 | 76.75±0.96 |
| Solarize | 48.72±1.55 | 63.58±1.12 | 70.14±0.73 | 73.94±0.02 |
| SolarizeAdd | 52.48±2.06 | 64.95±0.34 | 69.17±0.20 | 71.91±0.49 |
| TranslateX | 41.41±1.39 | 54.97±0.94 | 64.88±0.39 | 70.03±0.68 |
| TranslateY | 55.12±1.12 | 68.23±0.53 | 73.44±0.02 | 76.98±0.48 |

## B.4    ADDITIONAL RESULTS COMPARED TO RANDAUGMENT.

Since CAMPAL locate feasible augmentations guided by their strength, we also compare CAMPAL with RandAugment (Cubuk et al., 2020). To show the effectiveness of a separate control on unlabeled/labeled data in CAMPAL, we trained RandAugment on the labeled data within each AL cycle, then applied the optimized augmentation to both the labeled pool and unlabeled pool. As shown in Table 6, CAMPAL shows better performance than the RandAugment, indicating the superiority of the separate control. It should be noted that RandAugment is originally designed for training over full labeled data, but is obliged to be conducted over the labeled pool with limited samples under the AL setting. Therefore, directly adopting RandAugment to AL is infeasible, since it can be heavily biased towards limited labeled data, contributing little to the distribution enrichment on unlabeled data.

## B.5    ABLATION STUDIES OVER TYPES OF AUGMENTATIONS

**The impact of each single-image augmentation operator on CAMPAL.**    To further dive into the impact of the contribution of augmentations, we also provide the results when each augmentation is separately applied to CAMPAL with different strengths, shown in Table 7 on CIFAR-10 with CAMPAL$_{\text{Entropy}}^{\text{DENSITY}}$. We can see the impact of different types of single-image augmentations varies. An interesting observation is that different augmentation operator does not contribute equally at the different AL cycles. For example, Sharpness performs better than Rotate when $\mathcal{N}_L = 500$, but underperforms Rotate when $\mathcal{N}_L = 2000$. It reveals a sophisticated mechanism of the benefit of these augmentation operators on AL. However, the profound theory behind why data augmentation works have not been fully revealed to date, making it difficult to principally pick up the best optimal augmentation type. Hence, we naively adopt a simple strategy that uniformly selects and stacks these operators to enjoy their mixed benefits to AL.

Table 8: Test accuracy of CAMPAL when integrated with different combinations of single-image augmentations and the MixUp.

| Augmentations | | Entropy | LC | Margin | Core-set | BADGE |
|---|---|---|---|---|---|---|
| Single | MixUp | | | | | |
| ✓ | | 75.87±0.32 | 76.04±0.41 | 77.61±0.91 | 76.62±0.50 | 77.78±0.45 |
| | ✓ | 75.41±0.25 | 74.89±0.64 | 77.76±0.48 | 75.21±0.32 | 75.69±0.62 |
| ✓ | ✓ | **76.90±0.76** | **76.82±0.62** | **78.70±0.58** | **78.20±0.28** | **79.71±0.51** |

Table 9: Test accuracy of CAMPAL when integrated with different combinations of single-image augmentations and the MixUp.

| Coefficients | | Entropy | LC | Margin | Core-set | BADGE |
|---|---|---|---|---|---|---|
| $\lambda_1$ | $\lambda_2$ | | | | | |
| 1.0 | 0 | 75.87±0.32 | 73.36±2.25 | 70.15±2.19 | 76.62±0.50 | 77.78±0.45 |
| 1.0 | 0.5 | 75.62±0.76 | 75.89±0.73 | 77.20±0.52 | 79.23±0.51 | 77.07±0.38 |
| 0 | 1.0 | 75.41±0.25 | 72.93±0.65 | 67.69±0.72 | 73.99±0.03 | 75.69±0.62 |
| 0.5 | 1.0 | 75.95±0.62 | 76.55±0.62 | 76.25±0.60 | 75.21±0.32 | 74.88±0.14 |
| 1.0 | 1.0 | 76.90±0.76 | 76.82±0.62 | 78.70±0.58 | 78.20±0.28 | 79.71±0.51 |

**Effect of single-image augmentations and mix-up.**    To prove the efficacy of including both single-image augmentations of image-mixing into one query batch, we further explore the effect of these two kinds of augmentations separately. To verify this, we conduct experiments over two variants of CAMPAL that only use one type of augmentations, i.e. single-image augmentations and MixUp. The tests are performed by the ResNet-18 model with 4% (2000) data from CIFAR-10. For fairness, when only one kind of augmentation is used, we generate 15 augmented samples of this type. In Table 8, we can see a consistent performance boost when using both kinds of augmentations over Entropy ($\Delta$ 1.03), LC ($\Delta$ 0.78), Margin ($\Delta$ 0.94), Coreset ($\Delta$ 1.58), and BADGE ($\Delta$ 1.93). In conclusion, an integration of both single-image augmentations and image-mixing better unleashes the potential information of each sample than they separately do.

**Effect of $\lambda_1, \lambda_2$ in the virtual loss term.**    To optimize $m_l$, i.e. the strength for augmentations performed over labeled samples, we use $\lambda_1, \lambda_2$ to trade off the impact of single-image augmentations and image mixing. We dive deeper into this scheme by applying different combinations of $\lambda_1, \lambda_2$, shown in Table 9. Specifically, the experiment is conducted on the following versions: 1) CAMPAL$_{\text{Entropy}}^{\text{MIN}}$; 2) CAMPAL$_{\text{LC}}^{\text{MIN}}$; 3) CAMPAL$_{\text{Margin}}^{\text{MIN}}$; 4) CAMPAL$_{\text{Coreset}}^{\text{STANDARD}}$; 5) CAMPAL$_{\text{BADGE}}^{\text{STANDARD}}$.

### B.6    FURTHER EXTENSION: AUGMENTATIONS VS. UNSUPERVISED TRAINING

Recall that several studies tried to involve unlabeled samples in training auxiliary networks to assist querying(Sinha et al., 2019; Zhang et al., 2020; Kim et al., 2021a; Caramalau et al., 2021), which inevitably brings high computational costs. We claim that data augmentations are sufficient to enforce the acquisition process without much extra cost over unsupervised training. To verify this, we compare the running time and performance of augmentation-based strategies and those utilizing extra unsupervised architectures, shown in Table 10. We can see that augmentation-based methods with the best performance consistently outperform other strategies when becoming computationally efficient. Since active learning usually faces the problem of heavy computational cost in acquisitions, data augmentation may serve as an effective tool for both boosting the speed and performance at once. More importantly, this thought restricts the training process merely over labeled data, thus reducing the need for numerous unlabeled data in AL and making AL paradigms more applicable. We also adopt augmentations for labeled samples for methods with unsupervised representations.

Table 10: Comparison of the averaged test accuracy and the run-time of a single AL cycle over CIFAR-10. The run-time is calculated as the ratio to Random Sampling. Bold indicates the best performance of different data scales within each category.

| Method | | $\mathcal{N}_L = 500$ | $\mathcal{N}_L = 1,000$ | $\mathcal{N}_L = 1,500$ | $\mathcal{N}_L = 2,000$ | Time |
|---|---|---|---|---|---|---|
| Random | | 38.54±2.28 | 49.77±3.08 | 58.61±2.75 | 61.49±2.06 | 1 |
| Entropy | | 39.80±1.60 | 55.43±1.71 | 60.76±2.64 | 65.95±1.36 | 1.03 |
| LC | | 38.50±1.10 | 53.83±2.71 | 59.74±2.12 | 66.97±1.87 | 1.01 |
| Margin | | 40.03±2.49 | 54.22±2.47 | 62.61±1.91 | 66.76±1.77 | 1.07 |
| Core-set | | 43.42±2.09 | 53.54±2.74 | 62.00±1.44 | 66.90±0.93 | 1.33 |
| BADGE | | 44.18±2.09 | 55.97±1.57 | 62.40±2.15 | 67.03±0.62 | 1.28 |
| Unsupervised Representation | TA-VAAL | 61.72±0.47 | 66.67±0.92 | 70.53±0.50 | 74.41±0.70 | 5.82 |
| | SRAAL | 60.53±0.89 | 67.08±0.28 | 71.02±0.66 | 75.05±0.15 | 6.04 |
| | CoreGCN | 56.03±1.73 | 59.81±1.31 | 65.19±1.49 | 69.61±2.34 | 2.73 |
| CAMPAL-based Augmentation | Entropy | 62.78±1.33 | 69.34±1.35 | 71.84±1.35 | 76.90±0.76 | 5.13 |
| | LC | 61.89±0.80 | 69.06±1.00 | 73.49±0.92 | 76.82±0.62 | 5.08 |
| | Margin | 65.46±0.63 | 72.77±0.55 | 75.96±0.85 | 78.70±0.58 | 5.10 |
| | Core-set | 62.59±0.89 | 71.55±0.29 | 75.69±0.62 | 78.20±0.28 | 5.42 |
| | BADGE | **66.40±1.01** | **73.48±0.42** | **77.38±0.53** | **79.71±0.51** | 5.54 |

### B.7 ADDTIONAL RESULTS FOR ABLATION STUDIES OVER STRENGTHS

**Heatmap visualization for Core-set and BADGE.** We also provide different combinations of strengths on $\text{CAMPAL}_{\text{Coreset}}^{\text{STANDARD}}$ and $\text{CAMPAL}_{\text{BADGE}}^{\text{STANDARD}}$, which have similar phenomena as described in Section 3.3 and shown in Figure 7. Specifically, augmentations over unlabeled samples tend to produce better results as strength increases and produce the best results mostly with strength 4, indicating that drastic augmentations help induce a stronger acquisition. In contrast, augmentations over labeled samples produce the best results mostly with a strength of 2, fitting with the conclusion in Section 3.2 that weak augmentations for labeled samples are better at boosting the classifier. These phenomena show the difference between the impacts of augmentations over training and acquisition in active learning, and further guarantee the importance of a combinatorial scheme of weak and strong augmentations being strategically applied to labeled and unlabeled data.

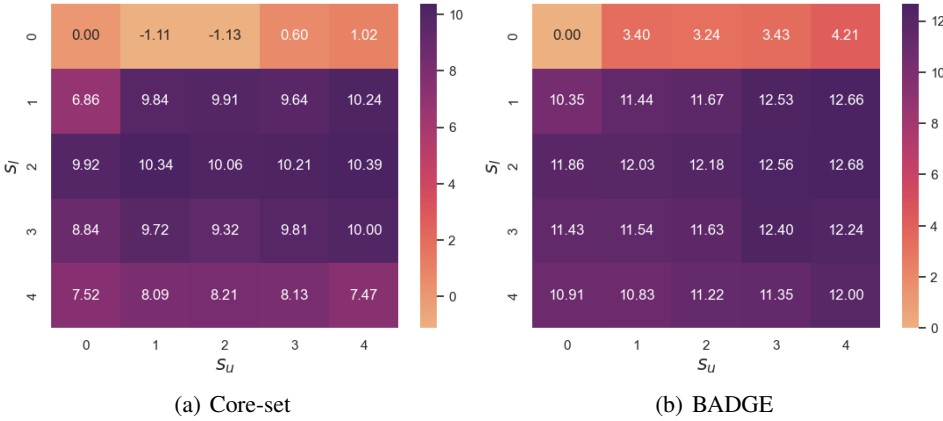

(a) Core-set                         (b) BADGE

Figure 7: A heatmap visualization of performance boost brought by augmentations of different strengths, when attached to the labeled pool and unlabeled pool. The experiments are performed over CIFAR-10 with 2,000 labeled samples and are conducted over Core-set and BADGE.

**Strength Change Across Different Cycles.** Since we show the strength on CIFAR-10 change across AL cycles for $\text{CAMPAL}_{\text{Entropy}}^{\text{MIN}}$ and $\text{CAMPAL}_{\text{BADGE}}^{\text{STANDARD}}$ in the main content, we also show other versions of strength change on CIFAR-10 in Figure 8. We can also see that larger $m_u$'s in comparison with $m_l$'s. We ran each experiment with 100 cycles and repeat them 5 times. In more

detail, the difference between $m_u, m_l$ is not clear in the early stages of active learning, since the model lacks the ability to uncover information from images.

## C PSEUDO CODE

We summarize the pseudo-code of our CAMPAL within one active learning cycle in Algorithm 1.

---

**Algorithm 1:** An active learning cycle for CAMPAL.

---

**Require :** Labeled data pool $\hat{\mathcal{D}}_L$, Unlabeled data pool $\mathcal{D}_U$, Model $f_\theta$.

$\theta \leftarrow \arg\min_\theta \frac{1}{|\mathcal{D}_L|} \sum_{x \in \mathcal{D}_L} \mathcal{L}\left(f_\theta(x), y\right)$;

$m_l = \arg\min_m \frac{1}{|\mathcal{D}_L|} \sum_{x_L \in \mathcal{D}_L} \mathcal{L}_f(x_L, m)$, where $L_f$ is shown in equation 2;

Generate an augmentation set $\mathcal{T}^{(m_l)}$ with strength $m_l$;

$\theta \leftarrow \arg\min_\theta \frac{1}{|\mathcal{T}^{(m_{lab})}(\mathcal{D}_L)|} \sum_{x \in \mathcal{T}^{(m_l)}(\mathcal{D}_L)} \mathcal{L}\left(f_\theta(x), y\right)$;

$m_u = \arg\max_m \sum_{x_U \in \mathcal{D}_U} \min\{\mathbb{H}(\tilde{x}_U) \,|\, \tilde{x}_U \in \mathcal{T}^{(m)}(x_U),\, f_\theta(\tilde{x}_U) = f_\theta(x_U)\}$;

Generate an augmentation set $\mathcal{T}^{(m_u)}$ with strength $m_u$;

Deduce the enhanced acquisition $h_{acq}$ with $\mathcal{T}^{(m_u)}$ and $f_\theta$ as shown in Section 2.3;

Select optimal sample batch $\mathcal{Q}$ according to $h_{acq}$;

$\mathcal{D}_U \leftarrow \mathcal{D}_U - \mathcal{Q}$;

$\mathcal{D}_L \leftarrow \mathcal{D}_L \cup \mathcal{Q}$.

---

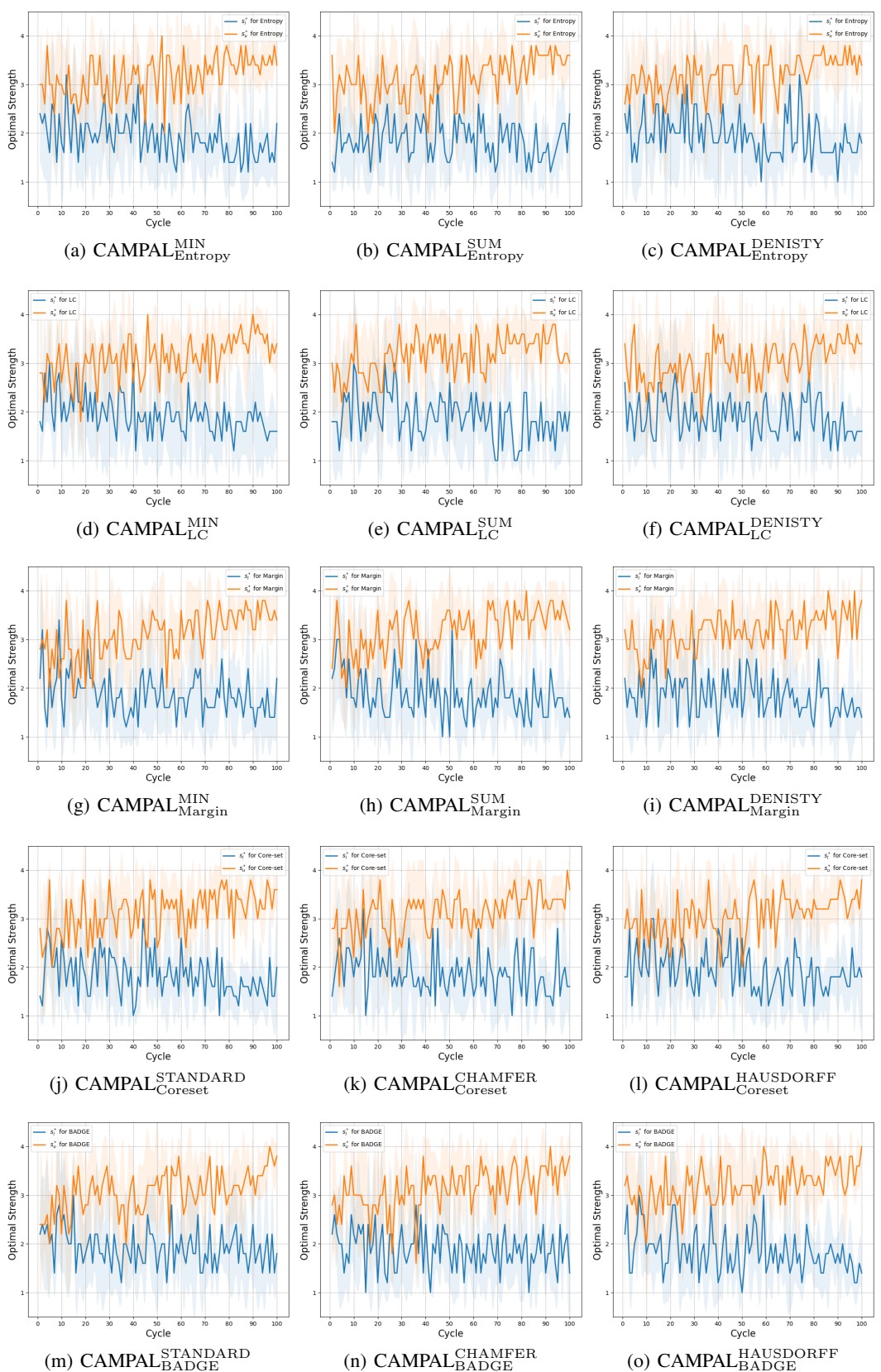

Figure 8: The average optimal strength $m_l, m_u$ across different AL cycles on CIFAR-10 with different instantiated versions for CAMPAL.

