# OpenReview forum: "Active Learning with Controllable Augmentation Induced Acquisition"
_ICLR.cc/2023/Conference — Submitted to ICLR 2023_

### Official Review · Reviewer_mJxF · 2022-10-22

**Confidence:** 5
**Clarity, Quality, Novelty And Reproducibility:** The paper is generally well-written a…
**Correctness:** 4
**Technical Novelty And Significance:** 3
**Empirical Novelty And Significance:** 2
**Recommendation:** 5

**Strength And Weaknesses:**

Strengths -

1. I like the motivation mentioned in the paper about separating the role of augmentations over the labeled and unlabeld set.
2. The ablation study does show that indeed the strengths of augmentation over unlabeled set is higher than labeled set, which justifies the reasoning of the paper.
3. I also like that the paper shows the comparison against having a fixed augmentation strength, where it is shown that certain combination of fixed augmentation strength can be non-optimal and hence the proposed approach of optimizing for the augmentation strength is better.

Weakness -

1. My main weakness of the paper is that since it is also considering augmentations over the unlabeled set, a fairer comparison would be the semi-supervised learning approaches such as [1] which achieve much superior results with much fewer samples. The existing Active Learning approaches do not play with the unlabeled samples generally and thus their setup is much simpler and unfair to compare with other semi-supervised learning setups. In contrast, the proposed approach here explicitly optimizes the augmentation strength over unlabeled samples, making their setup complex. If indeed an ML practitioner wants to use unlabeled data for their model, they instead can rely on using semi-supervised learning instead of the proposed approach here which involves multiple steps. The paper's worth would be improved much more if the authors show that their method can be added on top of the semi-supervised approaches such as [1] too.

2. While I like the comparison done in the appendix with RandAugment, it seems that the proposed approach is not very better than RandAugment (only 1-2%). Further, the authors optimize for RandAugment over labeled samples. What happens if we instead use the RandAugment policy that's been optimized over larger datasets such as ImageNet and then use the same policy across all the datasets mentioned in the paper?

3. How do the authors select their best performing CAMPL, as there are different search axes, which active learning to use and also which aggregation mode to use during including the augmentations in the Active learning approach. Do the authors have a separate validation set and then select the best variant by its performance over the validation set? I could not see any reference to the validation set in the paper. In particular when comparing against RandAugment, is the validation set same for learning the best RandAugment policy and for selecting the best CAMPL approach?

4. Since the proposed approach involves multiple steps, I believe the time taken for this would be much larger than other active learning approaches, however, I could not find a mention of it. Can authors also compare the time taken for their approach against the baseline Active learning approaches?

References -
[1]FixMatch: Simplifying Semi-Supervised Learning with Consistency and Confidence. Sohn et al.

**Summary Of The Paper:**

The paper proposes a new active learning approach by taking into consideration augmentations during the Active Learning acquisition phase. The paper argues that the existing active learning baselines do not effectively utilize unlabeled samples, particularly during the early stages of acquisition. To fix this, the authors propose to use augmentations both for training over the labeled samples as well as for acquisition over the unlabeled samples. Importantly, the way augmentations are used differs for the labeled and the unlabeled samples. For labeled samples, we want to have label-preserving augmentations that ensure consistency in the model's output across augmentations. However, for unlabeled samples we want the augmentations to be informative enough to enrich the distribution of the unlabeled data, making it easier for the active learning strategy to acquire the best samples for labeling, in other words, the augmentations that provide the maximum least information across the unlabeled samples. The paper thus optimizes for the strength of the augmentations (or the number of augmentations) separately over the labeled samples and the unlabeled samples using the above-mentioned criteria. Once the augmentation strengths are chosen, the authors propose different strategies in which the augmentation over the unlabeled samples can be used for the acquisition, both for score-based and representation-based acquisition functions. Via extensive study, the authors show that their proposed approach performs better than the different Active Learning baselines over different datasets and across different labeled data sizes. In addition the paper also proposes a theoretical justification behind their approach.

**Summary Of The Review:**

To summarize, while I agree that the paper's idea is good and novel I have my concerns with the experimental setup. The paper would do great if the proposed approach could be applied over semi-supervised approaches also. I also have some concerns over the comparison with a fixed augmentation strategy, RandAugment.

I would be willing to update my ratings if my above concerns are addressed.

---

> ### Author Response · Authors · 2022-11-14
> **Response to R3(2)**
>
> **Q2: About RandAugment(RA)**
>
> **A.** This is a very good point, thank you! As you requested, we also conducted additional experiments with RA policy optimized on several datasets. We do whole-heartedly agree with the question the reviewer raised: how does our proposed strength/magnitude data augmentation compare with a pre-trained and then transferred one such as RandAug?
>
> First, we supplemented a set of experiments on CIFAR-10 and observe their full-cycle performances. The following table shows comparative results on CIFAR-10 with fewer samples, showing the performance boost is significant with 200 annotated samples (6.30%). Since CAMPAL optimizes augmentation strengths for labeled/unlabeled samples at each cycle towards their own objectives, it does not rely on data quantity and is much more flexible than RandAugment which relies on fixed parameters. The performance difference seems to amplify especially with fewer labeled samples at the early stages for AL.
>
> | Method              | $N_L=200$ | $N_L=500$ | $N_L=1,000$ | $N_L=2,000$ |
> | ------------------- | --------- | --------- | ----------- | ----------- |
> | Ent w. RA(CIFAR-10) | 33.96     | 55.55     | 68.17       | 76.83       |
> | Ent w. RA(ImageNet) | 34.39     | 53.14     | 66.90       | 77.85       |
> | CAMPAL              | 40.26     | 58.06     | 71.40       | 78.89       |
>
> Second, we argue that with approaches like RandAugment, there are normally two ways of specific implementation:
>
> - For one thing, one can always obtain a new RandAugment policy on the provided task associated with the dataset. However, we argue this is not quite realistic in the active learning setup where the labeled data is limited and every step of label acquisition extracts a toll. What is more adverse is that the original RandAugment approach evidently requires a large number of validation samples (e.g. 10,000 samples on CIFAR-10) to be involved.
> - For the other, following reviewers’ request, one can obtain a policy from an external source and then transfer it to the current setting. While this is functionally feasible, we do observe no strong performance gain from the above table. While this way of transferring a pre-trained RandAugment policy is promising, we believe it is also pivotal in terms of how to transfer/fine-tune it in the course of an AL setting. This may have gone beyond the scope of this paper.
>
>
>
> **Q3: About selecting the best CAMPAL approach.**
>
> **A.** Yes, we adopted a validation set for model selection, as it is a common practice in the ML community. Empirically, our validation set consists of 20% samples split from the original training set. Lastly, RandAugment is optimized according to this validation set, and the validation set is the same across different experiments.
>
>
>
> **Q4: About the computational cost.**
>
> **A.** In our work, the extra computational costs are two-fold: 1) extra training cost brought by labeled augmentations; 2) extra evaluation cost brought by optimizing strengths and aggregation. Both of them just add a constant multiplier when calculating complexity compared to their original baselines. For example, a max-entropy sampling algorithm has a time complexity of $O(U\log K)$, where $U$ denotes the number of unlabeled samples and $K$ denotes the number of samples to query. Then $\mbox{CAMPAL}_{\rm Entropy}$ has a time complexity of $O(U logK + mU)$, where $m$ is the number of augmentations and $O(mU)$ is brought by aggregating information scores and selecting strength. Table 10 in Appendix B.7 shows the time taken for an AL cycle compared to methods with auxiliary architectures. We also present some of the results in the following table:
>
> | Method                        | Performance        | Time |
> | ----------------------------- | ------------------ | ---- |
> | Random                        | 61.49$\pm$2.06     | 1    |
> | Entropy                       | 65.95$\pm$1.36     | 1.03 |
> | BADGE                         | 67.03$\pm$0.62     | 1.28 |
> | TA-VAAL                       | 74.41$\pm$0.70     | 5.82 |
> | SRAAL                         | 75.05$\pm$0.15     | 6.04 |
> | $\mbox{CAMPAL}_{\rm Entropy}$ | 76.90$\pm$0.76     | 5.13 |
> | $\mbox{CAMPAL}_{\rm BADGE}$   | **79.71$\pm$0.51** | 5.54 |
>
> The run-time is calculated as the ratio to the Random acquisition. Each experiment is performed on CIFAR-10. Since the time taken with different aggregations seldom varies, we omit the aggregation superscript over CAMPAL. Clearly, the running time of our CAMPAL method is at the same scale as the baselines.
>
> Perhaps most importantly, as we consider the very down-to-earth deployment of an AL framework, the running time is not usually the prioritized metric for the practitioner cohorts to consider. This is primarily because every cycle of data querying would require a human annotator (also dubbed the oracle) to get involved and annotate the data. This procedure would normally cost way more than the machine running time.

---

> ### Author Response · Authors · 2022-11-14
> **Response to R3(1)**
>
> Thank you for your comments! Answers to your questions follow.
>
> **Q1: About the usage of the unlabeled pool in active learning.**
>
> **A:** The answer to this question very much overlaps with Q1 for reviewer ev91 :). The main discrimination between AL and SSL lies in whether they interfere with the training process on classifiers. In the typical setting of AL, the unlabeled samples do not interfere with the training process, when SSL train the classifier over both of them. The high performance of the current SSL like the FixMatch method largely relies on enrolling the unlabeled data into the training procedure, e.g. by self-training or consistency loss. In more detail, active learning has different paradigms, objectives, and applications scenarios from that of semi-supervised learning as follows:
>
> 1. **Paradigm Difference.** The SSL normally requires all the unlabeled data to possess during training, e.g. by self-training or consistency-loss. By contrast, AL methods would only draft a few samples cycle-by-cycle from the unlabeled pool and use an oracle to provide the associative labels. Besides the small batch drafted at every cycle for labeling, AL does not involve the unlabeled data during training.
>
> 2. **Goal and Objective Difference.** Namely, the ultimate goal of the AL approach is to lower the cost of data annotation. It attempts to form an annotation query from the unlabeled data pool to attain optimal return reflected by the model performance gain. Put another way, the final outcome of active learning is composed of two parts --- a trained model as well as a larger labeled data pool which presumably costs much less than random selection. By contrast, the SSL approaches may only yield a model.
>
> 3. **Suitable Applications Differ.** In certain scenarios where the unlabeled data has a certain barrier or cost to obtain --- such as edge devices [1,2], when facing privacy problems [3,4], or when acquiring unlabeled data is quite expensive [5,6] --- AL is the best suitable paradigm than other learning forms including the SSL.
>
>    In that follows, we may take drug discovery as an example. Normally, the primary goal of the drug-discovery task is to identify promising candidates within a fixed candidate pool. In this scenario, the unlabeled data here is not trivial to apply within the normal SSL paradigm because these candidate pools may exhibit no profound structure at the distribution level. For instance, in peptide discovery, the candidate unlabeled pool amounts to a full permutation of the basic amino-acid elements. Here, the unlabeled data (also the searching candidate pool) lies on a huge multi-dimensional grid with no explicit distributional pattern. Hence, in this scenario, AL is a natural solution while SSL cannot suffice. Other subjects that present uninformative unlabeled features like molecular generation [7], and chemical process optimization [8] also integrate AL into their optimization procedure.
>
> In our work, unlabeled is merely employed for evaluation. **We never involve unlabeled samples in model training or use any SSL techniques or tricks (e.g. pseudo labeling, consistency loss).** The optimization is performed to search the strength, which is much more efficient than training on unlabeled data. *In effect, there is also a plethora of active learning researches that focus on the standard AL task in top-tiered AI conferences like ICML, NeurIPS, and CVPR*. Regarding this, we believe it is not appropriate to compare the AL algorithms to the leading SSL frameworks.
>
> [1] Chen, Cheng, Yi Li, and Yiming Sun. "Online Active Regression." *International Conference on Machine Learning*. PMLR, 2022.
>
> [2] Ahmed, Lulwa, et al. "Active learning based federated learning for waste and natural disaster image classification." *IEEE Access* 8 (2020): 208518-208531.
>
> [3] Lenczner, Gaston, et al. "DIAL: Deep Interactive and Active Learning for Semantic Segmentation in Remote Sensing." *IEEE Journal of Selected Topics in Applied Earth Observations and Remote Sensing* 15 (2022): 3376-3389.
>
> [4] Desai, Bimbisar, et al. "Rapid discovery of a novel series of Abl kinase inhibitors by application of an integrated microfluidic synthesis and screening platform." *Journal of medicinal chemistry* 56.7 (2013): 3033-3047.
>
> [5] Xin, Rui, et al. "Active-Learning-Based Generative Design for the Discovery of Wide-Band-Gap Materials." *The Journal of Physical Chemistry C* 125.29 (2021): 16118-16128.
>
> [6] Shields, Benjamin J., et al. "Bayesian reaction optimization as a tool for chemical synthesis." *Nature* 590.7844 (2021): 89-96.
>
> [7] Xin, Rui, et al. "Active-Learning-Based Generative Design for the Discovery of Wide-Band-Gap Materials." *The Journal of Physical Chemistry C* 125.29 (2021): 16118-16128.
>
> [8] Shields, Benjamin J., et al. "Bayesian reaction optimization as a tool for chemical synthesis." *Nature* 590.7844 (2021): 89-96.

---

### Official Review · Reviewer_VeMh · 2022-10-25

**Confidence:** 4
**Correctness:** 4
**Technical Novelty And Significance:** 3
**Empirical Novelty And Significance:** 4
**Recommendation:** 8

**Clarity, Quality, Novelty And Reproducibility:**

The paper is clearly rewritten with high quality. The novelty is also guaranteed with its newly-developed augmentation flows constructed distinctly on labeled and unlabeled data towards their own objectives. The reproducibility is also guaranteed with the code given.

**Strength And Weaknesses:**

The paper proposes a new augmentation-acquisition framework for active learning, namely CAMPAL, to better enhance active learning with data augmentations. The author empirically and theoretically verified the effectiveness of CAMPAL. The point that the augmentation flows impacts AL differently on labeled and unlabeled data pool is also interesting.
The strengths are:
1.	The proposed method is novel and well-motivated, and is clearly written in this paper.
2.	This paper fully investigates an overlooked research problem in active learning, and illustrates its importance with significant performance boosts.
3.	The authors also provide a theoretical guarantee of the proposed method.
The weaknesses are:
1.	This emphasizes the efficacy of CAMPAL when labeled data are scarce. How about the efficacy of CAMPAL in a normal setting, i.e. with 10%~40% labeled samples from the dataset?
2.	How do you define the image similarity $\rm{sim}(x,\tilde{x})$ in the score-based acquisition in Section 2.3? Please elaborate on it.
3.	As shown in Figure 3 and Table 3, the performance boost on CIFAR-100 brought by CAMPAL is not as significant as that on SVHN, and CIFAR-10. It would be better to give out an explanation for this phenomenon.
4.	In equation (2) in Section 2.2, the term $\frac{\lambda_{2}}{\mathcal{T}^{(s)}(x)}$ is confusing, maybe it is $\frac{\lambda_2}{|\mathcal{T}_{\rm mix}^{(s)}(x)|}$.


**Summary Of The Paper:**

This paper proposes a method, CAMPAL, for better enhancing active learning with data augmentations. To achieve this, they provide a strength controlling mechanism for augmentations integrated into active learning, as well as a newly developed augmentation-acquisition procedure. The author also provides strong empirical results on several benchmarks, together with theoretical guarantees. The paper brings an overlooked research interest for active learning, fully investigating the role of data augmentations when integrating them into active learning.

**Summary Of The Review:**

The paper proposes a new augmentation-acquisition framework for active learning, CAMPAL, to better enhance active learning with data augmentations. The proposed method is quite effective on several benchmarks, especially with scarce labeled data at the first few active learning cycles. The data augmentation is important and effective in active learning, but very few works have been devoted to this problem. Therefore, this paper is great to fully investigating the impacts of data augmentations on active learning.

---

> ### Author Response · Authors · 2022-11-14
> **Response to R2**
>
> Thank you very much for the comments and suggestions! We are happy you enjoyed the paper.
>
> **Q1: The experimental results with more labeled samples.**
>
> | Method  | $N_L$=2,000        | $N_L$=5,000        | $N_L$=10,000       | $N_L$=20,000       |
> | ------- | ------------------ | ------------------ | ------------------ | ------------------ |
> | Random  | 61.49$\pm$2.06     | 76.59$\pm$0.77     | 79.87$\pm$0.81     | 85.56$\pm$0.16     |
> | Entropy | 65.95$\pm$1.36     | 81.73$\pm$0.54     | 86.12$\pm$0.90     | 88.46$\pm$0.34     |
> | BADGE   | 67.03$\pm$0.62     | 83.60$\pm$0.76     | 88.62$\pm$0.58     | 90.77$\pm$0.41     |
> | CAMPAL* | **80.37$\pm$0.86** | **85.80$\pm$0.91** | **90.92$\pm$0.84** | **93.89$\pm$0.47** |
>
> **A.** The table above shows a comparison between different AL strategies when more samples are queried and labeled on CIFAR-10, with each experiment repeated 3 times. We can see that CAMPAL also brings a consistent performance gain across different data scales. But this boost tends to decline as labeled data size grows. It is because the growth of data volume makes the underlying data distribution denser, thus weakening the role of data augmentation for enriching data distribution.
>
> **Q2: The definition of image similarity $\mbox{sim}(x,\tilde{x})$ in Section 2.3.**
>
> **A.** We apologize for this confusion! $\mbox{sim}(x,\tilde{x})=\exp(-0.5*d(f_{\rm emb}(x), f_{\rm emb}(\tilde{x}))$, where $f_{\rm emb}(x)$ denotes the feature embedding provided by the penultimate layer of our ResNet-18 architecture, with $d(\cdot,\cdot)$ denotes the euclidean distance. This follows the information-density evaluation setup provided by settles et al [1].
>
> [1] B. Settles and M. Craven. An analysis of active learning strategies for sequence labeling tasks. In Proceedings of the Conference on Empirical Methods in Natural Language Processing (EMNLP), pages 1069–1078. ACL Press, 2008.
>
> **Q3: The performance boost on CIFAR-100.**
>
> **A.** The performance boost on CIFAR-100 is not as significant as those on CIFAR-10, and SVHN, but it is indeed a notable improvement. In detail, our proposed CAMPAL brings a performance boost of 3.99% with 10,000 annotated samples compared to previous baselines, when current works only improve the accuracy by 1%~2% [2] or even fail to outperform them [3,4].
>
> [2] Kwanyoung Kim, Dongwon Park, Kwang In Kim, and Se Young Chun. Task-aware variational
> adversarial active learning. In Proceedings of the IEEE/CVF Conference on Computer Vision and
> Pattern Recognition, pp. 8166–8175, 2021a.
>
> [3] Parvaneh, Amin κ.ά. ‘Active Learning by Feature Mixing’. *Proceedings of the IEEE/CVF Conference on Computer Vision and Pattern Recognition (CVPR)*. N.p., 2022. 12237–12246.
>
> [4] Yoon-Yeong Kim, Kyungwoo Song, JoonHo Jang, and Il-Chul Moon. Lada: Look-ahead data
> acquisition via augmentation for deep active learning. Advances in Neural Information Processing
> Systems, 34:22919–22930, 2021b.
>
> **Q4: typo: equation (2).**
>
> Thank you! We have carefully polished our manuscripts and corrected all the typos.

---

> > ### Comment · Reviewer_VeMh · 2022-11-22
> > **Responses to the rebuttal**
> >
> > I’d like to thank the authors for their detailed responses. After reading them, my questions have all been answered and addressed. I’m pretty happy with that and will keep my original score.

---

### Official Review · Reviewer_ev91 · 2022-10-30

**Confidence:** 3
**Correctness:** 3
**Technical Novelty And Significance:** 3
**Empirical Novelty And Significance:** 2
**Recommendation:** 3

**Clarity, Quality, Novelty And Reproducibility:**

I find the paper writing is fairly hand-waving --- numerous rigorous definitions and technical details. But, most importantly, the theoretical explanation seems disconnected from the proposed algorithm.

**Strength And Weaknesses:**

Strength:
1. The proposed method applies to a wide range of augmentation methods and acquisition methods.
2. The authors clearly explain the heuristics in the proposed algorithm.

Weakness:
1. My primary concern is about the comparison against semi-supervised learning. Given the same quota of labeled samples, active learning should outperform semi-supervised learning --- while active learning (AL) can actively select the training subset, semi-supervised learning (SSL) passively takes the training subset. However, I find the proposed algorithm significantly underperforms the MixMatch, the benchmarking method in SSL. For example, given 2000 samples, CutMix achieves an error rate of 7.03% for CIFAR-10 and 3.04% for SVHN (See Appendix B); in contrast, the proposed algorithm only gets 19.63% for CIFAR-10 and 14.34% for SVHN (See Table 1). It raises a natural question of why people would like to use AL instead of SSL.
2. The technical descriptions in Section 2.2 are somewhat hand-waving. The authors did not describe the math definitions of different types of augmentations. Also, it is unclear how to solve the optimization problems in (1) and (2). Whether these problems are solved in closed form or iteratively? Are they solved repeatedly once the labeled set is expanded or once at the beginning of the algorithm? Minor: I guess the denominator in (2) actually means the cardinality of the augmentations.
3. The theoretical explanations are disconnected from the actual algorithm. The paper does not justify why the proposed algorithm meets the assumptions/conditions in theory. For example, why the augmentations used in the paper is "moderately weak" (Assumption 2)? What does it mean by "properly selected" in Theorem 4? Without rigorous development, the theory will not shed light on the algorithm.

**Summary Of The Paper:**

The paper proposes a heuristic pipeline to improve existing active learning methods using controllable augmentation.

**Summary Of The Review:**

Overall, I think the paper lacks technical rigor (see Weakness for details) --- unfortunately, I cannot recommend acceptance.

---

> ### Author Response · Authors · 2022-11-14
> **Response to R1(2)**
>
> **Q2: About the definitions of augmentations and the optimization procedure.**
>
> **A:** We apologize for this confusion! In our revised manuscript, we have added a detailed definition of our augmentation techniques. Please refer to Appendix B.3 for details.
>
> For the optimization procedure, we solve both (1) and (2) in a closed form by traversing different strengths. More specifically, we optimize two global variants that quantify the overall strengths for augmentations applied to labeled/unlabeled data, whose search space is extremely small. This small search space enables us to traverse all the strengths and select the optimal one. (1) is optimized before the model training stage at each AL cycle, while (2) is optimized before the querying stage at each AL cycle. With the processes above, it is straight-forward to optimize strengths by simply traversing.
>
>
>
> **Q3: Minor**
>
> **A:** Our bad! The denominator $|\mathcal{T}_{mix}^{(s)}(x)|$ in Eq. (2) indeed means the cardinality of augmentations, we have corrected this typo error.
>
>
>
> **Q4: The relationship between our theory and method.**
>
> **A:** Sorry for this confusion. Let us provide more details in understanding the relationship between our theory and our algorithms.
>
> CAMPAL automatically selects suitable strengths for augmentations performed over labeled/unlabeled augmentations separately. Through such a controlled augmentation strategy, CAMPAL is able to find proper augmentation policies guided by strengths for both labeled and unlabeled data. Empirically, we do observe that the labeled data prefer weak augmentations for ensuring prediction accuracy, while unlabeled data has more freedom in evaluating informativeness with more aggressive augmentations. This observational result is provided in the Experiment section. Hence, our theories are offered mainly to interpret such phenomenon. We hope to provide more solid reasoning alongside the empirical findings to back the validity of our approach.
>
> - For labeled samples, the "moderately weak" statement from Assumption 2 guarantees the correctness of the error bound in Eq. (3), and this error bound measures the reliability of this classifier. Since we rely on a fully-trained classifier to induce dependable informativeness evaluation as we describe in Section 2.3, it is important to ensure that this error bound holds. When augmentations for labeled samples are too strong, the overlapping between $C_i'$s should be also considered, making the form for this bound more complicated so as to increase and leads to an undependable classifier. Thus we use the "moderately weak" statement to avoid infeasible augmentations and ensure the reliability of informativeness induced by the classifier.
> - For unlabeled samples, we aim to find the most effective sample that maximizes the error bound reduction Eq. (4). The error bound reduction corresponds to the informativeness of queried samples, which is exactly what we try to enhance in Section 2.2 and our experiments. Specifically, weak augmentations make the $P_{L+1}$ and $\Delta m$ small, which in turn causes an insufficient reduction on the error bound, meaning the informativeness of the queried sample is not enhanced enough. Therefore, this theory leads to that the unlabeled samples should be enabled stronger augmentations than labeled samples, consistent with our experimental results. An intuitive explanation for this is that the stronger augmentation applied to the unlabeled data aims at enhancing informativeness evaluation, rather than improving the performance of the classifier at the current cycle. To conclude, this theory explains why AL enables stronger augmentations performed over unlabeled data.
>
> We have also added more details for the optimal strength of unlabeled/labeled augmentations in Figure 8 and Appendix B.7, with an intuitive illustration shown in Figure 6 in Appendix A. In conclusion, both our theoretical derivations and experimental results show that AL is better enhanced by DA with a combinatorial scheme of weak and strong augmentations applied to labeled and unlabeled data.

---

> ### Author Response · Authors · 2022-11-14
> **Response to R1(1)**
>
> Thanks for the comments! The answer to your questions follows.
>
> **Q1: About active learning(AL) and semi-supervised learning(SSL).**
>
> **A:** Noted, our approach has established a new record for active learning on several standardized benchmarks, surpassing all prior AL methods by significant margins. However, it is true that our scores have not yet reached the level of Mix-Match as well as perhaps other SSL approaches. In that regard, we respectfully disagree with the reviewer on comparing AL approaches with the ones in the SSL regime.
>
> For one thing, active learning, as a long-standing paradigm in machine learning, evolves on a very separate and parallel line as semi-supervised learning. To put it straight, if we’d compare the results of every paper in the AL community with Mix-Match, we are afraid that none of the papers could have been published if judging from the results alone.
>
> For another, we argue that active learning and semi-supervised learning possess notable differences in their fundamental setups, and both have values for the researchers to get devoted:
>
> 1. **Paradigm Difference.** The SSL normally requires all the unlabeled data to possess during training, e.g. by self-training or consistency-loss. By contrast, AL methods would only draft a few samples cycle-by-cycle from the unlabeled pool and use an oracle to provide the associative labels. Besides the small batch drafted at every cycle for labeling, AL does not involve the unlabeled data during training.
>
> 2. **Goal and Objective Difference.** Namely, the ultimate goal of the AL approach is to lower the cost of data annotation. It attempts to form an annotation query from the unlabeled data pool to attain optimal return reflected by the model performance gain. Put another way, the final outcome of active learning is composed of two parts --- a trained model as well as a larger labeled data pool which presumably costs much less than random selection. By contrast, the SSL approaches may only yield a model.
>
> 3. **Suitable Applications Differ.** In certain scenarios where the unlabeled data has a certain barrier or cost to obtain --- such as edge devices [1,2], when facing privacy problems [3,4], or when acquiring unlabeled data is quite expensive [5,6] --- AL is the best suitable paradigm than other learning forms including the SSL.
>
>    In that follows, we may take drug discovery as an example. Normally, the primary goal of the drug-discovery task is to identify promising candidates within a fixed candidate pool. In this scenario, the unlabeled data here is not trivial to apply within the normal SSL paradigm because these candidate pools may exhibit no profound structure at the distribution level. For instance, in peptide discovery, the candidate unlabeled pool amounts to be a full permutation of the basic amino-acid elements. Here, the unlabeled data (also the searching candidate pool) lies on a huge multi-dimensional grid with no explicit distributional pattern. Hence, in this scenario, AL is a natural solution while SSL cannot suffice. The community also evidently shows the importance of the AL methods. Desai et al. [7] found the best compound for Abl1 kinase inhibitor with 22 experiments (which might cost 270 experiments without AL). Other subjects that present uninformative unlabeled features like molecular generation [8], and chemical process optimization [9] also integrate AL into their optimization procedure.
>
>
>
> [1] Chen, Cheng, Yi Li, and Yiming Sun. "Online Active Regression." *International Conference on Machine Learning*. PMLR, 2022.
>
> [2] Liu, Sanmin, et al. "Online active learning for drifting data streams." *IEEE Transactions n Neural Networks and Learning Systems* (2021).
>
> [3] Ahmed, Lulwa, et al. "Active learning based federated learning for waste and natural disaster image classification." *IEEE Access* 8 (2020): 208518-208531.
>
> [4] Kelli, Vasiliki, et al. "IDS for industrial applications: a federated learning approach with active personalization." *Sensors* 21.20 (2021): 6743.
>
> [5] Lenczner, Gaston, et al. "DIAL: Deep Interactive and Active Learning for Semantic Segmentation in Remote Sensing." *IEEE Journal of Selected Topics in Applied Earth Observations and Remote Sensing* 15 (2022): 3376-3389.
>
> [6] Peng, Fengchao, et al. "Active learning for lane detection: A knowledge distillation approach." *Proceedings of the IEEE/CVF International Conference on Computer Vision*. 2021.
>
> [7] Desai, Bimbisar, et al. "Rapid discovery of a novel series of Abl kinase inhibitors by application of an integrated microfluidic synthesis and screening platform." *Journal of medicinal chemistry* 56.7 (2013): 3033-3047.
>
> [8] Xin, Rui, et al. "Active-Learning-Based Generative Design for the Discovery of Wide-Band-Gap Materials." *The Journal of Physical Chemistry C* 125.29 (2021): 16118-16128.
>
> [9] Shields, Benjamin J., et al. "Bayesian reaction optimization as a tool for chemical synthesis." *Nature* 590.7844 (2021): 89-96.

---

### Comment · Area_Chair_sCmi · 2022-11-26
**Following up on authors’ response and discussion**

Dear Reviewers,

Thank you very much again for performing this extremely valuable service to the ICLR community.

Please check the authors’ response and leave comments if you have not done it.

Best,

AC

---

### Decision · Program_Chairs · 2023-01-20

**Decision:**

Reject

**Justification For Why Not Higher Score:**

It is unclear why one should use the proposed active learning method instead of using existing semi-supervised ones.

**Justification For Why Not Lower Score:**

N/A

**Metareview: Summary, Strengths And Weaknesses:**

This paper suggests a new active learning (AL) pipeline named CAMPAL by introducing controllable data augmentation in the acquisition (for unlabeled data) and training (for labeled data) phase. For controllable data augmentation, the author optimizes the strength of the augmentations (i.e., the number of applied augmentations) over the labeled and unlabeled samples, respectively, with the proposed objective. Two reviewers are negative, and one reviewer is positive. The main concern of negative reviewers is that CAMPAL underperforms the semi-supervised learning (SSL) method (i.e., MixMatch) even though CAMPAL uses additional annotation query and optimization over the unlabeled dataset. The author argues that CAMPAL has a different setup and suitable applications compared to SSL approaches. However, considering additional annotation cost and optimization over an unlabeled dataset in CAMPAL, AC thinks that in many scenarios, ML practitioners given an unlabeled dataset would prefer SSL approaches, which show better performance than CAMPAL. As the authors mentioned that AC and SSL are two separate (or even somewhat irrelevant) learning paradigms, one can combine AC and SSL for even better performance, which AC thinks the authors should do to avoid the criticism of the inferior performance of AC to SSL. Reviewer mJxF indeed suggested to apply CAMPAL on top of SSL, but the authors did not provide the corresponding results. Overall, AC tends to recommend rejection.